# Identity Crisis: Memorization and Generalization under Extreme Overparameterization

**Chiyuan Zhang & Samy Bengio**
Google Research, Brain Team
Mountain View, CA 94043, USA
{chiyuan,bengio}@google.com

**Moritz Hardt**
University of California, Berkeley
Berkeley, CA 94720, USA
hardt@berkeley.edu

**Michael C. Mozer**
Google Research, Brain Team
Mountain View, CA 94043, USA
mcmozer@google.com

**Yoram Singer**
Princeton University
Princeton, NJ 08544, USA
y.s@princeton.edu

## Abstract

We study the interplay between memorization and generalization of overparameterized networks in the extreme case of a single training example and an identity-mapping task. We examine fully-connected and convolutional networks (FCN and CNN), both linear and nonlinear, initialized randomly and then trained to minimize the reconstruction error. The trained networks stereotypically take one of two forms: the constant function (*memorization*) and the identity function (*generalization*). We formally characterize generalization in single-layer FCNs and CNNs. We show empirically that different architectures exhibit strikingly different inductive biases. For example, CNNs of up to 10 layers are able to generalize from a single example, whereas FCNs cannot learn the identity function reliably from 60k examples. Deeper CNNs often fail, but nonetheless do astonishing work to memorize the training output: because CNN biases are location invariant, the model must progressively grow an output pattern from the image boundaries via the coordination of many layers. Our work helps to quantify and visualize the sensitivity of inductive biases to architectural choices such as depth, kernel width, and number of channels.

## 1 Introduction

The remarkable empirical success of deep neural networks is often attributed to the availability of large data sets for training. However, sample size does not provide a comprehensive rationale since complex models often outperform simple ones on a given data set, even when the model size exceeds the number of training examples.

What form of inductive bias leads to better generalization performance from highly overparameterized models? Numerous theoretical and empirical studies of inductive bias in deep learning have been conducted in recent years (Dziugaite & Roy, 2016; Kawaguchi et al., 2017; Bartlett et al., 2017; Neyshabur et al., 2017; Liang et al., 2017; Neyshabur et al., 2018; Arora et al., 2018; Zhou et al., 2019) but these postmortem analyses do not identify the root source of the bias.

One cult belief among researchers is that gradient-based optimization methods provide an implicit bias toward simple solutions (Neyshabur et al., 2014; Soudry et al., 2018; Shah et al., 2018; Arora et al., 2019). However, when a network is sufficiently large (e.g., the number of hidden units in each layer is polynomial in the input dimension and the number of training examples), then under some mild assumptions, gradient methods are guaranteed to fit the training set perfectly (Allen-Zhu et al., 2018b; Du et al., 2018a;b; Zou et al., 2018). These results do not distinguish a model trained on a data distribution with strong statistical regularities from one trained on the same inputs but with randomly shuffled labels. Although the former model might achieve good generalization, the latter can only memorize the training labels. Consequently, these analyses do not tell the whole story on the question of inductive bias.

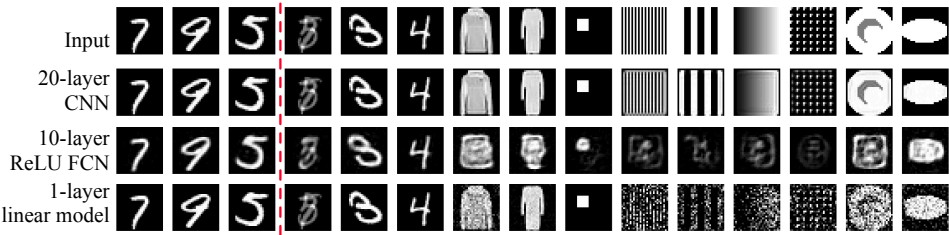

Figure 1: Predictions of three architectures trained on the *identity mapping* task with 60k MNIST examples. The red dashed line separates 3 training examples from the test examples.

Another line of research characterizes sufficient conditions on the input and label distribution that guarantee generalization from a trained network. These conditions range from linear separability (Brutzkus et al., 2018) to compact structures (Li & Liang, 2018). While very promising, this direction has thus far identified only structures that can be solved by linear or nearest neighbor classifiers over the original input space. The fact that in many applications deep neural networks significantly outperform these simpler models reveals a gap in our understanding of deep neural networks.

As a formal understanding of inductive bias in deep networks has been elusive, we conduct a novel exploration in a highly restrictive setting that admits visualization and quantification of inductive bias, allowing us to compare variations in architecture, optimization procedure, initialization scheme, and hyperparameters. The particular task we investigate is learning an *identity mapping* in a regression setting. The identity mapping is interesting for four reasons. First, it imposes a structural regularity between the input and output, the type of regularity that could in principle lead to systematic generalization (He et al., 2016; Hardt & Ma, 2017). Second, it requires that every input feature is transmitted to the output and thus provides a sensitive indicator of whether a model succeeds in passing activations (and gradients) between inputs and outputs. Third, conditional image generation is a popular task in the literature (e.g., Mirza & Osindero, 2014; Ledig et al., 2017); an identity mapping is the simplest form of such a generative process. Fourth, and perhaps most importantly, it admits detailed analysis and visualization of model behaviors and hidden representations.

Consider networks trained on the identity task with 60k MNIST digits. Although only digit images are presented during the training, one might expect the strong regularity of the task to lead to good generalization to images other than digits. Figure 1 compares three different architectures. The top row shows various input patterns, and the next three rows are outputs from a 20-layer convolutional net (CNN), a 10-layer fully connected net (FCN) with rectified-linear unit (ReLU) activation functions, and a 1-layer FCN. The 1-layer FCN amounts to a convex optimization problem with infinitely many solutions, however gradient decent converges to a unique closed-form solution. All nets perform well on the training set (first three columns) and transfer well to novel digits and digit blends (columns 4–6). Yet, outside of the hull of hand-printed digits, only the CNN discovers a reasonably good approximation to the identity function.

Figure 1 reflects architecture-specific inductive bias that persists even with 60k training examples. Despite this persistence, a model's intrinsic bias is more likely to be revealed with a smaller training set. In this paper, we push this argument to the limit by studying learning with a *single* training example. Although models are free to reveal their natural proclivities in this maximally overparameterized regime, our initial intuition was that a single example would be uninteresting as models would be algebraically equivalent to the constant function (e.g., via biases on output units). Further, it seemed inconceivable that inductive biases would be sufficiently strong to learn a mapping close to the identity. Unexpectedly, our experiments show that model behavior is subtle and architecture dependent. In a broad set of experiments, we highlight model characteristics—including depth, initialization, and hyperparameters—that determine where a model lands on the continuum between *memorization* (learning a constant function) and *generalization* (learning the identity function). The simplicity of the training scenario permits rich characterization of inductive biases.

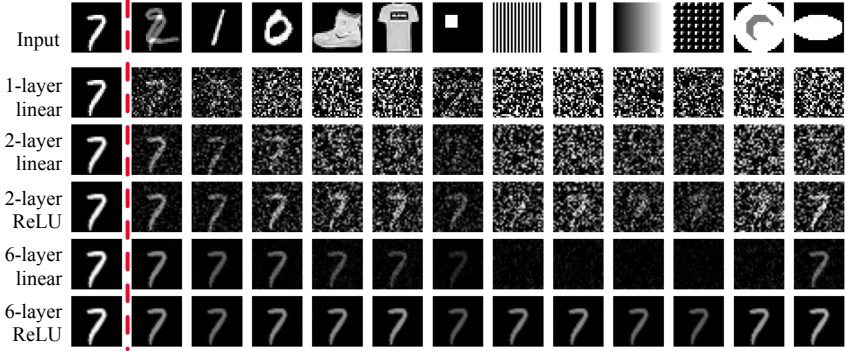

Figure 2: **Visualization of the outputs of fully connected networks trained on a single example.** The first row shows the single training example (7) and a set of evaluation images consisting of a linear combination of two digits, random digits from MNIST test set, random images from Fashion MNIST, and some algorithmically generated image patterns. Each row below indicates an architecture and the output from that architecture for a given input.

## 2 RELATED WORK

The consequences of overparameterized models in deep learning have been extensively studied in recently years, on the optimization landscape and convergence of SGD (Allen-Zhu et al., 2018b; Du et al., 2018a;b; Bassily et al., 2018; Zou et al., 2018; Oymak & Soltanolkotabi, 2018), as well as the generalization guarantees under stronger structural assumptions of the data (Li & Liang, 2018; Brutzkus et al., 2018; Allen-Zhu et al., 2018a). Another line of related work is the study of the implicit regularization effects of SGD on training overparameterized models (Neyshabur et al., 2014; Zhang et al., 2017; Soudry et al., 2018; Shah et al., 2018; Arora et al., 2019).

The traits of memorization in learning are also explicitly studied from various perspectives such as prioritizing learning of simple patterns (Arpit et al., 2017) or perfect interpolation of the training set (Belkin et al., 2018; Feldman, 2019). More recently, coincidentally with the writing of this paper, Radhakrishnan et al. (2018) reported on the effects of the downsampling operator in convolutional auto-encoders on image memorization. Their empirical framework is similar to ours, fitting CNNs to the autoregression problem with few training examples. We focus on investigating the general inductive bias in the extreme overparameterization case, and study a broader range of network types without enforcing a bottleneck in the architectures.

## 3 EXPERIMENTS

We explore a progression of models: linear convex models, linear non-convex models (with multiple linear layers), fully-connected multilayered architectures with nonlinearities, and finally the case of greatest practical importance, fully convolutional networks. In all architectures we study, we ensure that there is a simple realization of the identity function (see Appendix B). We train networks by minimizing the *mean squared error* using standard gradient descent.

### 3.1 FULLY CONNECTED NETWORKS

Figure 2 shows examples of predictions from multi-layer fully connected networks. Infinitely many solutions exist for all models under this extreme over-parameterization, and the figure shows that all the models fit the training example perfectly. However, on new test examples, contrasting behaviors are observed between shallow and deep networks. In particular, deeper models bias toward predicting a constant output, whereas shallower networks tend to predict random white noises on unseen inputs. The random predictions can be characterized as follows (proof in Appendix C).

**Theorem 1.** *A one-layer fully connected network, when trained with gradient descent on a single training example $\hat{x}$, converges to a solution that makes the following prediction on a test example $x$:*

$$f(x) = \Pi_{\parallel}(x) + \mathbf{R}\Pi_{\perp}(x),\tag{1}$$

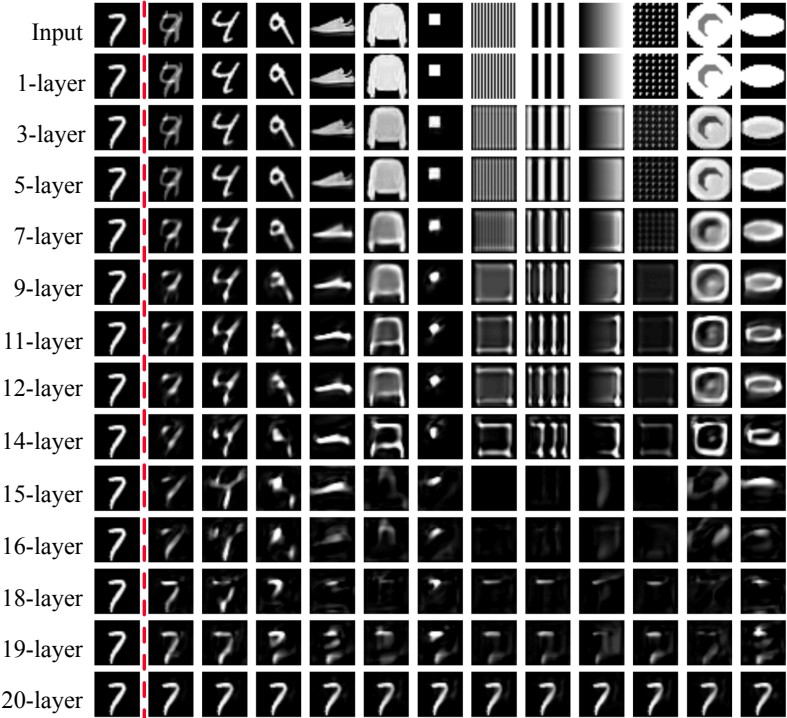

Figure 3: **Visualization of predictions from CNNs trained on a single example.** The first row shows the single training example (7) and a set of test inputs. Each row below shows the output of a CNN whose depth is indicated to the left of the row. The hidden layers of the CNN consist of $5 \times 5$ convolution filters organized as 128 channels.

*where $x = \Pi_{\parallel}(x) + \Pi_{\perp}(x)$ decomposes $x$ into orthogonal components that are parallel and perpendicular to the training example $\hat{x}$, respective. $\mathbf{R}$ is a random matrix from the network initialization, independent of the training data.*

For test examples similar to the training example—i.e., where $\Pi_{\parallel}(x)$ dominates $\Pi_{\perp}(x)$—the outputs resemble the training output; on the other hand, for test examples that are not highly correlated to the training example, $\Pi_{\perp}(x)$ dominates and the outputs looks like white noise due to the random projection by $\mathbf{R}$. The behavior can be empirically verified from the second row in Figure 2. Specifically, the first test example is a mixture of the training and an unseen test example, and the corresponding output is a mixture of white noise and the training output. For the remaining test examples, the outputs appear random.

Although Theorem 1 characterizes only the 1-layer linear case, the empirical results in Figure 2 suggest that shallow (2 layer) networks tend to have this inductive bias. However, this inductive bias does not miraculously obtain good generalization: the trained model fails to learn either the identity or the constant function. Specifically, it predicts well in the vicinity (measured by correlations) of the training example $\hat{x}$, but further away its predictions are random. In particular, when the test example $x$ is orthogonal to $\hat{x}$, the prediction is completely random. In deeper (6 layer) nets, the deviations from the identity function take on a quite different characteristic form.

Interestingly, deeper *linear* networks behave more like deeper ReLU networks, with a strong bias towards a constant function that maps any input to the single training output. A multilayer *linear* network with no hidden-layer bottleneck has essentially the same representational power as a 1-layer linear network, but gradient descent produces different learning dynamics that alter the inductive biases. See Appendix E for more results and analysis on FCNs.

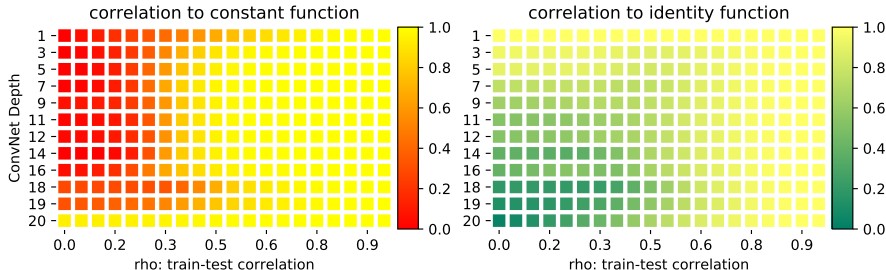

Figure 4: **Predictions of CNNs on test examples at different angles to the training image.** The horizontal axis shows the train-test correlation, while the vertical axis indicate the number of hidden layers for the CNNs being evaluated. The heatmap shows the similarity (measured in correlation) between the model prediction and the reference function (the constant or the identity function).

### 3.2 CONVOLUTIONAL NETWORKS

We next study the inductive bias of convolutional neural networks with ReLU activation functions. Figure 3 shows predictions on various test patterns obtained by training CNNs of varying depths. Compared to FCNs, CNNs have strong structural constraints that limit the receptive field of each neuron to a spatially local neighborhood and the weights are tied and being used across the spatial array. These two constraints match the structure of the identity target function. (See Appendix B.3 for an example of constructing the identity function with CNNs.) Similar to the fully connected case, for one-layer CNN, we can bound the error as follows (proof in Appendix D).

**Theorem 2.** *A one-layer convolutional neural network can learn the identity map from a single training example with the mean squared error over all output pixels bounded as*

$$MSE \leq \tilde{\mathcal{O}} \left( \frac{m(m/C - r)}{C} \right) \tag{2}$$

*where $m$ is the number of network parameters, $C$ is the number of channels in the image, and $r \leq m/C$ is the rank of the subspace formed by the span of the local input patches.*

The error grows with $m$, the number of parameters in the network. For example, learning CNNs with larger receptive field sizes will be harder. Even though the bound seems to decrease with more (input and output) channels in the image, note that the number of channels $C$ also contributes to the number of parameters ($m = K_H K_W C^2$), so there is a trade-off. Unlike typical generalization bound that decays with number of i.i.d. training examples, we have only one training example here, and the key quantity that reduces the bound is the rank $r$ of the subspace formed by the local image patches. The size of the training image implicitly affects bounds as larger image generates more image patches. Note the rank $r$ also heavily depends on the contents of the training image. For example, simply padding the image with zeros on all boundaries will not reduce the error bound. With enough linearly independent image patches, the subspace becomes full rank $r = m/C$, and learning of the global identity map is guaranteed.

The theorem guarantees only the one-layer case. Empirically—as shown in Figure 3—CNNs with depth up-to-5 layers learn a fairly accurate approximation to the identity function, with the exception of a few artifacts at the boundaries. For a quantitative evaluation, we measure the performance by calculating the correlation (See Appendix J for the results in MSE.) to two reference functions: the identity function and the constant function that maps every input to the training point $\hat{x}$. To examine how a model's response varies with similarity to the training image, we generate test images having correlation $\rho \in [0, 1]$ to the training image by: (1) sampling an image with random pixels, (2) adding $\alpha \hat{x}$ to the image, picking $\alpha$ such that the correlation with $\hat{x}$ is $\rho$; (3) renormalizing the image to be of the same norm as $\hat{x}$. For $\rho = 0$, the test images are orthogonal to $\hat{x}$, whereas for $\rho = 1$, the test images equal $\hat{x}$. The results for CNNs of different depths are shown in Figure 4. The quantitative findings are consistent with the visualizations: shallow CNNs are able to learn the identity function from only one training example; very deep CNNs bias towards the constant function; and CNNs of intermediate depth correlate well with neither the identity nor the constant function. However,

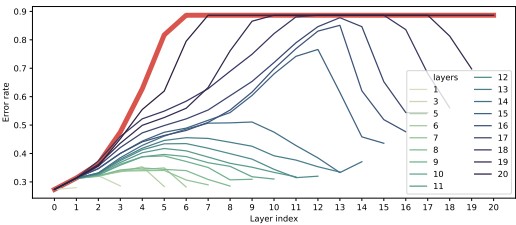

Figure 5: **Illustration of the collapse of predictive power as function of layer's depth.** Error rate is measured by using the representations computed at each layer to a simple averaging based classifier on the MNIST test set. The error rate at each layer is plotted for a number of trained CNNs of different depth. The thick red line shows the curve of an *untrained* 20-layer CNN for reference.

unlike FCNs that produce white-noise-like predictions, from Figure 3 CNNs of intermediate depth behave like edge detectors.

To evaluate how much information is lost in the intermediate layers, we use the following simple criterion to assess the representation in each layer. We feed each image in the MNIST dataset through a network trained on our single example. We collect the representations at a given layer and perform a simple similarity-weighted classification. For each example $x_i$ from the MNIST test set, we predict its class as a weighted average of the (one hot) label vector of each example $x_j$ from the MNIST training set, where the weight is the inner-product of $x_i$ and $x_j$.

This metric does not quantify how much information is preserved as the image representation propagates through the layers, because the representation could still be maintained yet not captured by the simple correlation-weighted classifier. It nonetheless provides a simple metric for exploring the (identity) mapping: using the input image as the baseline, if a layer represents the identity function, then the representation at that layer would obtain a similar error rate when using the input representation; on the other hand, if a layer degenerates into the constant function, then the corresponding representation would have error rate close to random guessing of the label. The results are plotted in Figure 5. The error curve for a randomly initialized 20-layer CNN is shown as reference: at random initialization, the smoothing effect renders the representations beyond the sixth layer unuseful for the averaging-based classifier. After training, the concave nonmonotonicity in the curves indicates loss and then recovery of the information present in the input. Trained networks try to recover the washed out intermediate layer representations as means to link the input and the output layer. However, if the depth is too large, the network tries to infer input-output relations using partial information, resulting in models that behave like edge detectors. Finally, for the case of 20 layers, the curve shows that the bottom few layers do get small improvements in error rate comparing to random initialization, but the big gap between the input and output layer drives the network to learn the constant function instead. On a first sight, this deems to underscore a vanishing gradient problem, but Figure 1 reveals that given a sufficient number of training examples, a 20-layer CNN *can* still learn the identity map. See also Appendix G for further discussions on vanishing gradients. Since CNNs preserve spatial structure, we can also visualize information loss in the intermediate layers. The visualization results, described in Appendix F, are consistent with the aforementioned observations.

### 3.3 ROBUSTNESS TO CHANGES IN INPUT SCALE

Whether a relatively shallow CNN learns the identity function from a single training example or a relatively deep CNN learns the constant function, both outcomes reflect an inductive bias because the training objective never explicitly mandates the model to learn one structure or another. The spatial structure of CNNs enables additional analyses of the encoding induced by the learned function. In particular, we examined how changing the size of the input by scaling the dimensions of the spatial map affects the model predictions. Figure 6 depicts the predictions of a 5-layer CNN trained on a $28 \times 28$ image and tested on $7 \times 7$ and $112 \times 112$ images. Although the learned identity map generally holds up against a larger-than-trained input, the identity map is disturbed on smaller-than-trained inputs. See Appendix H.1 for a comprehensive description of the results. Note that CNNs are capable of encoding the identity function for arbitrary input and filter sizes (Appendix B.3).

Figure 7 shows the predictions on the rescaled input patterns of Figure 6 by a 20-layer CNN that has learned the constant function on a $28 \times 28$ image. The learned constant map holds up over a smaller range of input sizes than the learned identity map in a 5-layer CNN. It is nonetheless interesting to see smooth changes as the input size increases to reveal the network's own notion of "7", clearly

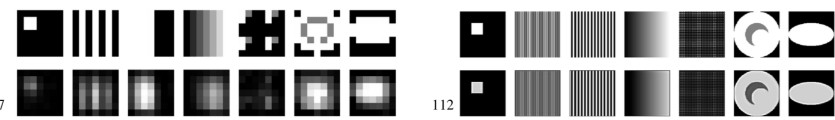

Figure 6: **Visualization of a 5-layer CNN on test images of different sizes**. The two subfigures show the results on $7 \times 7$ inputs and $112 \times 112$ inputs, respectively.

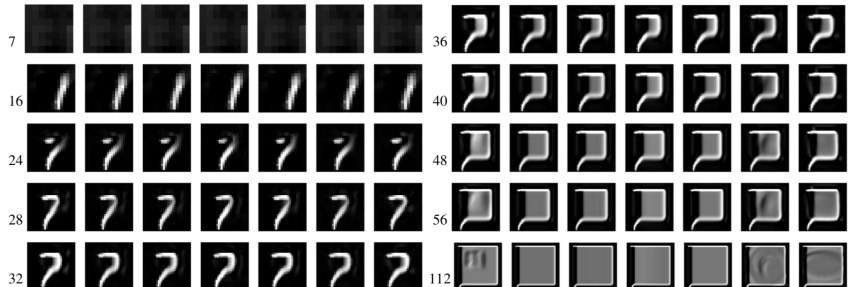

Figure 7: **Visualization of the predictions of a 20-layer CNN on test images of different sizes (indicated by the number on each row).** The input patterns are the same as in Figure 6 (constructed in different resolutions), which are not shown for brevity.

defined with respect to the corners of the input map (Figure 21 in Appendix H reveals interesting details on how the patterns progressively grow from image boundaries in intermediate layers). We performed additional experiments to directly feed test images to the upper subnet, which reveals more about the generative process by which the net synthesizes a constant output (Appendix H).

### 3.4 VARYING OTHER FACTORS DURING TRAINING

We studied a variety of common hyperparameters, including image dimensions, convolutional filter dimensions, number of kernels, weight initialization scheme, and the choice of gradient-based optimizer. With highly overparameterized networks, training converges to zero error for a wide range of hyperparameters. Within the models that all perform optimally on the training set, we now address how the particular hyperparameter settings affect the inductive bias. We briefly present some of the most interesting observations here; please refer to Appendix I for the full results and analyses.

**Size of training image.** Figure 8 shows, for varying training-image size, the mean correlation to the constant and the identity function at different depths within the network. The training examples are resized versions of the same image. The bias toward the constant function at a given depth increases with smaller training images. This finding makes sense considering that smaller images provide fewer pixel-to-pixel mapping constraints, which is aligned with Theorem 2.

**Size of convolution filters.** Figure 9 illustrates the inductive bias with convolution filter size varying from $5 \times 5$ to $57 \times 57$. Predictions become blurrier as the filter size grows. With extremely large filters that cover the entire input array, CNNs exhibit a strong bias towards the constant function.

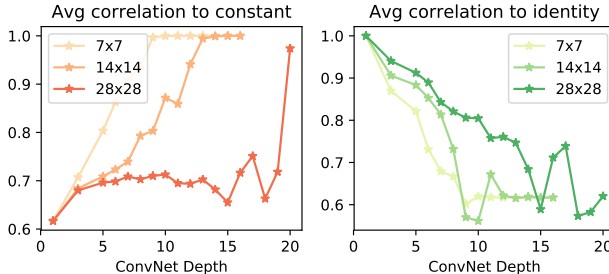

Figure 8: **Comparing bias towards constant and identity when trained with different image sizes.** The x-axis is the depth of the CNNs, while the y-axis is the mean correlation (average of each row from the heatmaps like in Figure 4). Each curve corresponds to training with a different image size.

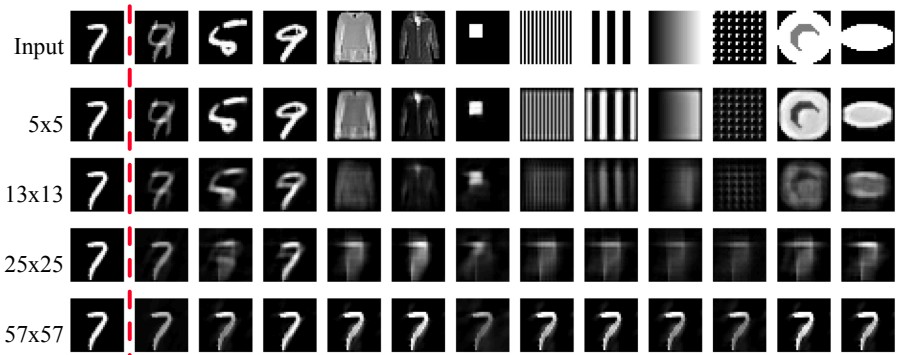

Figure 9: **Visualizing the predictions from 5-layer CNNs with various filter sizes.** The first row shows the single training example (7) and a set of test inputs. Each row below shows the output of a CNN whose filter size is indicated by the number on the left. See Appendix I for more results.

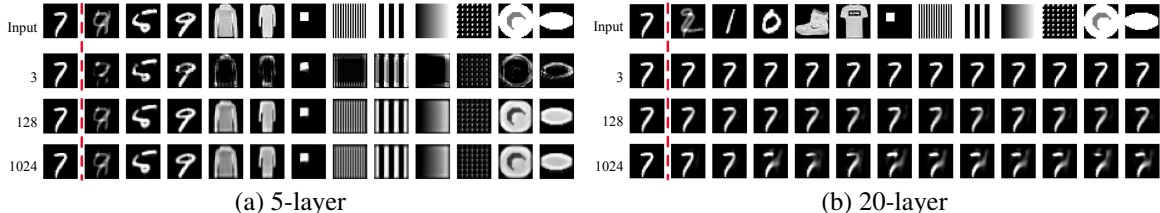

| (a) 5-layer | (b) 20-layer |

Figure 10: **Visualization of the predictions from CNNs for various number of hidden channels.** The first row shows a single training example (7) and a set of test inputs. Each row below shows the output of CNNs differing in the number of channels. This number is indicated on the left of each row; for intermediate layers, the number specifies both the input and output channel count.

Because the training inputs are of size $28 \times 28$, a $29 \times 29$ filter size allows each neuron to see at least half of the spatial domain of the previous layer, assuming large boundary padding of the inputs. With $57 \times 57$ filters centered at any location within the image, each neuron sees the entire previous layer. This is also consistent with Theorem 2, in which the error bound deteriorates as the filter sizes increases. Note even with a large filter, CNNs do *not* perform the same elementary computation as FCNs because the (shared) convolution filter is repeatedly applied throughout the spatial domain.

**Number of channels.** Appendix B.3 shows that in principle two channels suffice to encode the identity function for gray-scale inputs via a straightforward construction. However, in practice the outcome of training may be quite different depending on the channel count, as shown in Figure 10(a). On the one hand, the aggressively overparameterized network with 1024 channels ($\sim$25M parameters per middle convolution layer) does not seem to suffer from overfitting. On the other hand, the results with three channels often lose content in the image center. The problem is not underfitting as the network reconstructs the training image (first column) correctly.

A potential reason why 3-channel CNNs do not learn the identity map well might be poor initialization when there are very few channels (Frankle & Carbin, 2019). This issue is demonstrated in Figure 29 of Appendix I. Our study of training with different initialization schemes indeed confirms that the random initial conditions have a big impact on the inductive bias (Appendix I).

Figure 10(b) shows the case for 20-layer CNNs. Surprisingly, having an order of magnitude more feature channels in every layer does not seem to help much at making the information flow through layers, as the network still learns to ignore the inputs and construct a constant output.

## 4  CONCLUSIONS

We presented an systematic study of the extreme case of overparameterization when learning from a single example. We investigated the interplay between memorization and generalization in deep

neural networks. By restricting the learning task to the identity function, we sidestepped issues such as the underlying optimal Bayes error of the problem and the approximation error of the hypothesis classes. This choice also facilitated rich visualization and intuitive interpretation of the trained models. Under this setup, we investigated gradient-based learning procedures with explicit memorization-generalization characterization. Our results indicate that different architectures exhibit vastly different inductive bias towards memorization and generalization.

For future work, we plan to extend the study to other domains and neural network architectures, like natural language processing and recurrent neural networks, and aim for more qualitative relationship between the inductive bias and various architecture configurations.

ACKNOWLEDGMENTS

We would like to thank Kunal Talwar, Hanie Sedghi, and Rong Ge for helpful discussions.

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

## A  Experiment details and hyper-parameters

We specify the experiment setups and the hyper-parameters here. Unless otherwise specified in each study (e.g. when we explicitly vary the number of convolution channels), all the hyper-parameters are set according to the default values here.

The main study is done with the MNIST dataset. It consists of grayscale images of hand written digits of size $28 \times 28$. For training, we randomly sample one digit from the training set (a digit '7') with a fixed random seed. For testing, we use random images from the test set of MNIST and Fashion-MNIST, as well as algorithmically generated structured patterns and random images. The training and test images are all normalized by mapping the pixel values in $\{0, 1, \dots, 255\}$ to $[0, 1]$, and then standardize with the mean $0.1307$ and standard deviation $0.3081$ originally calculated on the (full) MNIST training set.

The models are trained by minimizing the mean squared error (MSE) loss with a vanilla SGD (base learning rate 0.01 and momentum 0.9). The learning rate is scheduled as stagewise constant that decays with a factor of 0.2 at the 30%, 60% and 80% of the total training steps (2,000,000). No weight decay is applied during training.

For neural network architectures, the rectified linear unit (ReLU) activation is used for both fully connected networks (FCNs) and convolutional networks (CNNs). The input and output dimensions are decided by the data. The hidden dimensions for the FCNs are 2,048 by default. The CNNs use $5 \times 5$ kernels with stride 1 and padding 2, so that the geometry does not change after each convolution layer.

## B  Representation of the identity function using deep networks

In this section, we provide explicit constructions on how common types of neural networks can represent the identity function. Those constructions are only proof for that the models in our study have the capacity to represent the target function. There are many different ways to construct the identity map for each network architecture, but we try to provide the most straightforward and explicit constructions. However, during our experiments, even when the SGD learns (approximately) the identity function, there is no evidence suggesting that it is encoding the functions in similar ways as described here. We put some mild constraints (e.g. no "bottleneck" in the hidden dimensions) to allow more straightforward realization of the identity function, but this by no means asserts that networks violating those constraints cannot encode the identity function.

### B.1  Linear models

For a one-layer linear network $f(x) = Wx$, where $W \in \mathbb{R}^{d \times d}$, setting $W$ to the identity matrix will realize the identity function. For a multi-layer linear network $f(x) = (\prod_\ell W_\ell)x$, we need to require that all the hidden dimensions are not smaller than the input dimension. In this case, a simple concrete construction is to set each $W_\ell$ to an identity matrix.

### B.2  Multi-layer ReLU networks

The ReLU activation function $\sigma(\cdot) = \max(0, \cdot)$ discards all the negative values. There are many ways one can encode the negative values and recover it after ReLU. We provide a simple approach that uses hidden dimensions twice the input dimension. Consider a ReLU network with one hidden layer $f(x) = W_2\sigma(W_1x)$, where $W_2 \in \mathbb{R}^{d \times 2d}, W_1 \in \mathbb{R}^{2d \times d}$. The idea is to store the positive and negative part of $x$ separately, and then re-construct. This can be achieved by setting

$$W_1 = \begin{pmatrix} I_d \\ -I_d \end{pmatrix}, \quad W_2 = (I_d \quad -I_d)$$

where $I_d$ is the $d$-dimensional identity matrix. For the case of more than two layers, we can use the bottom layer to split the positive and negative part, and the top layer to merge them back. All the intermediate layers can be set to $2d$-dimensional identity matrix. Since the bottom layer encode all the responsives in non-negative values, the ReLU in the middle layers will pass through.

### B.3 Convolutional Networks

In particular, we consider 2D convolutional networks for data with the structure of multi-channel images. A mini-batch of data is usually formatted as a four-dimensional tensor of the shape $B \times C \times H \times W$, where $B$ is the batch size, $C$ the number of channels (e.g. RGB or feature channels for intermediate layer representations), $H$ and $W$ are image height and width, respectively. A convolutional layer (ignoring the bias term) is parameterized with another four-dimensional tensor of the shape $\bar{C} \times C \times K_H \times K_W$, where $\bar{C}$ is the number of output feature channels, $K_H$ and $K_W$ are convolutional kernel height and width, respectively. The convolutional kernel is applied at local $K_H \times K_W$ patches of the input tensor, with optional padding and striding.

For one convolution layer to represent the identity function, we can use only the center slice of the kernel tensor and set all the other values to zero. Note it is very rare to use even numbers as kernel size, in which case the "center" of the kernel tensor is not well defined. When the kernel size is odd, we can set

$$W_{\bar{c}chw} = \begin{cases} 1 & \bar{c} = c, \ h = \lfloor K_H/2 \rfloor, \ w = \lfloor K_W/2 \rfloor \\ 0 & \text{otherwise} \end{cases}$$

By using only the center of the kernel, we essentially simulate a $1 \times 1$ convolution, and encode a local identity function for each (multi-channel) pixel.

For multi-layer convolutional networks with ReLU activation functions, the same idea as in multi-layer fully-connected networks can be applied. Specifically, we ask for twice as many channels as the input channels for the hidden layers. At the bottom layer, separately the positive and negative part of the inputs, and reconstruct them at the top layer.

## C  Proof of Theorem 1

Consider the 1-layer linear model $f_W(x) = Wx$, where $x \in \mathbb{R}^d$ and $W \in \mathbb{R}^{d \times d}$. Let $\hat{x}$ be the single training example. The training objective is to minimize the empirical risk $\hat{R} = 1/2 \|f_W(x) - x\|_2^2$. The optimization problem is convex and well understood. Due to overparameterization, the solution of the empirical risk minimization is not unique. However, given randomly initialized weights $W^0$, gradient descent obtains a unique global minimizer.

The gradient of the empirical risk is

$$\frac{\partial \hat{R}}{\partial W} = (W - I)\hat{x}\hat{x}^\top \tag{3}$$

Gradient descent with step sizes $\eta_t$ and initialization weights $W^0$ updates weights as

$$W^T = W^0 - \sum_{t=1}^T \eta_t(W^{t-1} - I)\hat{x}\hat{x}^\top$$
$$:= W^0 + u_T \hat{x}^\top$$

where $u_T \in \mathbb{R}^d$ is a vector decided via the accumulation in the optimization trajectory. Because of the form of the gradient, it is easy to see the solution found by gradient descent will always have such parameterization structure. Moreover, under this parameterization, a unique minimizer exists that solves the equation

$$\hat{x} = f_W(\hat{x}) = W^0 \hat{x} + u\hat{x}^\top \hat{x}$$

via

$$\hat{u} = \frac{(I - W^0)\hat{x}}{\|\hat{x}\|_2^2} \tag{4}$$

Therefore, the global minimizer can be written as

$$\hat{f}_W(x) = (W^0 + \hat{u}\hat{x}^\top)x$$
$$= \frac{\hat{x}^\top x}{\|\hat{x}\|_2^2} \cdot \hat{x} + W^0 \left( x - \frac{\hat{x}^\top x}{\|\hat{x}\|_2^2} \cdot \hat{x} \right)$$

For the one-layer network case, the optimization problem is convex. Under standard conditions in convex optimization, gradient descent will converge to the global minimizer shown above.

We can easily verify that the red term is exactly the projection of $x$ onto the training example $\hat{x}$, while the blue term is the residual projection onto the orthogonal subspace.

# D   PROOF OF THEOREM 2

**Lemma 1.** *Consider the linear model $f(x) = W^\top x$, where $x \in \mathbb{R}^D$, $W \in \mathbb{R}^{D \times d}$. Let the training set be $\{(x_1, y_1), \ldots (x_N, y_N)\}$. Assume the model is overparameterized ($N \leq d$) and there is no sample redundancy (rank of the data matrix is $N$). Fitting $f(\cdot)$ by optimizing the square loss with gradient descent on the training set converges to $\hat{f}(x) = \hat{W}^\top x$, where*

$$\hat{W} = W^0 + X^\top (XX^\top)^{-1} (Y - XW^0). \tag{5}$$

$W^0 \in \mathbb{R}^{D \times d}$ *is the random initialization of weights,* $X = (x_1, \ldots, x_N)^\top \in \mathbb{R}^{N \times D}$ *and* $Y = (y_1, \ldots, y_N)^\top \in \mathbb{R}^{N \times d}$ *are the matrices formed by the inputs and outputs, respectively.*

**Lemma 2.** *With the same notation of Lemma 1, assume the model is underparameterized ($N > d$) and there is no feature redundancy (rank of the data matrix is $d$). Fitting $f(\cdot)$ by optimizing the square loss with gradient descent on the training set converges to $\hat{f}(x) = \hat{W}^\top x$ where*

$$\hat{W} = \left(X^\top X\right)^{-1} X^\top Y, \tag{6}$$

*Proof of Theorem 2.* Let $\hat{x} \in \mathbb{R}^{H \times W \times C}$ be the single training image of size $H \times W$, and $C$ color channels. Let $K_H \times K_W$ be the convolution receptive field size, so the weights of convolution filters can be parameterized via a 4-dimensional tensor $\Theta \in \mathbb{R}^{K_H \times K_W \times C \times C}$. Let $\Xi$ be the collection of 2D coordinates of the local patches from the input image that the convolutional filter is applied to, and $\mathsf{P}_{ij}(\hat{x}) \in \mathbb{R}^{K_H \times K_W \times C}$ be the local patch of $\hat{x}$ centered at the coordinate $(i, j) \in \Xi$. Note for our case the input and output are of the same shape, so the convolution stride is one, and $|\Xi| = H \times W$.

The empirical risk of fitting $\hat{x}$ with a one-layer convolution model can be written as

$$\hat{R} = \frac{1}{2} \sum_{(i,j) \in \Xi} \sum_{k=1}^{C} \left(\langle \Theta_{:::k}, \mathsf{P}_{ij}(\hat{x}) \rangle - \hat{x}_{ijk}\right)^2, \tag{7}$$

where $\Theta_{:::k} \in \mathbb{R}^{K_H \times K_W \times C}$ is the subset of convolution weights corresponding to the $k$-th output channel, and $\hat{x}_{ijk}$ is the pixel value at coordinate $(i, j)$ and channel $k$.

CASE 1 — overparameterization: $|\Xi| \leq K_H \times K_W \times C \times C$. Assumme the patches are linearly independent[1], with slight abuse of notatons, we represent the empirical risk in matrix form with the

---

[1]When two patches are identical, or more generally, when there is a patch $(\tilde{i}, \tilde{j}) \in \Xi$, which can be written as a linear combination of other patches:

$$\exists \alpha_{ij}, s.t. \quad \mathsf{P}_{\tilde{i}\tilde{j}}(\hat{x}) = \sum_{(i,j) \neq (\tilde{i}, \tilde{j})} \alpha_{ij} \mathsf{P}_{ij}(\hat{x})$$

Note because each local patch contains the target pixel: $\hat{x}_{ijk} \in \mathsf{P}_{ij}(\hat{x}), \forall k \in [C]$, so for all $k \in [C]$, the same coefficients $\{\alpha_{ij}\}$ can be used to written $\hat{x}_{\tilde{i}\tilde{j}k}$ as the same linear combination of the other (corresponding) pixels. Therefore, we can find a linearly independent basis for the patches and rewrite the fitting objectives with the basis.

following matrices[2]:

$$W := (\Theta_{:::1}, \ldots, \Theta_{:::C}) \in \mathbb{R}^{(K_H K_W C) \times C},$$

$$X := (\ldots, \mathsf{P}_{ij}(\hat{x}), \ldots)^\top \in \mathbb{R}^{|\Xi| \times (K_H K_W C)},$$

$$Y := \begin{pmatrix} \vdots & \vdots & \vdots \\ \hat{x}_{ij1} & \ldots & \hat{x}_{ijC} \\ \vdots & \vdots & \vdots \end{pmatrix} \in \mathbb{R}^{|\Xi| \times C},$$

and apply Lemma 1 to obtain $\hat{W}$. For a test example $x$, the prediction of the $(i, j)$-th (multi-channel) pixel is

$$\hat{W}^\top \mathsf{P}_{ij}(x) = \left( W^0 + X^\top \left( XX^\top \right)^{-1} \left( Y - XW^0 \right) \right)^\top \mathsf{P}_{ij}(x)$$

$$= W^{0\top} \left( I - X^\top \left( XX^\top \right)^{-1} X \right) \mathsf{P}_{ij}(x) + Y^\top (XX^\top)^{-1} X \mathsf{P}_{ij}(x)$$

Note we are learning the identity map, the $(i, j)$-th row of the learning target matrix $Y$ is the (multi-channel) pixel at the $(i, j)$-th coordinate of $\hat{x}$. This is exactly the center of the $(i, j)$-th patch $\mathsf{P}_{ij}(\hat{x})$, i.e. the $(i, j)$-th row of the matrix $X$. As a result, there is a linear projection matrix $\Lambda \in \mathbb{R}^{(K_H K_W C) \times C}$ that maps $X$ to $Y$: $X\Lambda = Y$. So

$$\hat{W}^\top P_{ij}(x) = W^{0\top} \Pi_X^\perp \mathsf{P}_{ij}(x) + \Lambda^\top \Pi_X^\| \mathsf{P}_{ij}(x)$$

where $\Pi_X^\| = X^\top (XX^\top)^{-1} X$ is the linear operator that projects a vector into the subspace spanned by the rows of $X$, and $\Pi_X^\perp = I - \Pi_X^\|$ is the operator projecting into the orthogonal subspace.

To compute the prediction errors, it is suffice to look at the errors at each $(i, j)$-th (multi-channel) pixel separately:

$$\text{Error}_{ij}^2 = \|\hat{f}(x)_{ij} - \Lambda^\top \mathsf{P}_{ij}(x)\|^2$$

$$= \left\| W^{0\top} \Pi_X^\perp \mathsf{P}_{ij}(x) + \Lambda^\top \Pi_X^\| \mathsf{P}_{ij}(x) - \Lambda^\top \mathsf{P}_{ij}(x) \right\|^2$$

$$= \left\| (W^0 - \Lambda)^\top \Pi_X^\perp \mathsf{P}_{ij}(x) \right\|^2$$

$$\leq \left( \left( \|W^{0\top} \Pi_X^\perp\| + \|\Lambda^\top \Pi_X^\perp\| \right) \|\mathsf{P}_{ij}(x)\| \right)^2.$$

Assume the absolute value of pixel values are bounded by $B$, then

$$\|\mathsf{P}_{ij}(x)\| \leq B\sqrt{K_H K_W C}. \tag{8}$$

The first two terms can be bounded according to the rank $\mathsf{r}$ of the subspace projection $\Pi_X^\|$. Let the nullity of the projection be $\mathsf{n} = K_H K_W C - \mathsf{r}$. Note the projection matrix can be decomposed as

$$\Pi_X^\perp = \sum_{k=1}^{\mathsf{n}} v_k v_k^\top$$

where $\{v_k\}_k$ is a orthonormal basis for the projection subspace.

$$\|\Lambda^\top \Pi_X^\perp\| = \left\| \Lambda^\top \sum_{k=1}^{\mathsf{n}} v_k v_k^\top \right\|$$

$$= \sqrt{\sum_{k=1}^{\mathsf{n}} \|\Lambda^\top v_k\|^2} \qquad \text{(Orthonormality of } \{v_k\}_k)$$

$$\leq \sqrt{\mathsf{n}\|\Lambda\|^2} \qquad \text{(sub-multiplicativity of the Frobenius norm)}$$

$$= \sqrt{\mathsf{n}C}. \tag{9}$$

---

[2]In particular, $W$ is used for both the "width" of the image and the parameters of the linear system. But the meaning should be clear from the context. The multi-dimensional tensors $\Theta_{:::k}$ and $\mathsf{P}_{ij}(\hat{x})$ are used interchangeably with their flatten column vector counterparts.

Similarly,

$$\left\| W^{0\top}\Pi_X^\perp \right\| = \sqrt{\sum_{k=1}^{\mathfrak{n}} \left\| W^{0\top} v_k \right\|^2}$$

Assume the entries of $W^0$ are initialized as i.i.d. Gaussians $\mathcal{N}(0, \sigma^2)$. Since $\{v_k\}_k$ are orthonormal vectors, for each $k$, let $\rho_k = \|W^{0\top}v_k\|^2/\sigma^2$, then $\rho_k$ is distributed according to $\chi^2$ distribution with $C$ degree of freedom. For any $1 < \zeta \leq M$, where $M$ is a constant chosen a priori, and for each $k = 1, \ldots, \mathfrak{n}$,

$$
\begin{aligned}
\mathsf{P}\left(\rho_k \geq \zeta C\right) &\leq e^{-\zeta tC}\mathbb{E}\left[e^{t\rho_k}\right], &&\text{(Markov's inequality)} \\
&= e^{-\zeta tC}(1 - 2t)^{-C/2}, \quad \forall t < 1/2, &&\text{(MGF of } \chi^2) \\
&= \exp\left(-C\left(\zeta t + 1/2\log(1 - 2t)\right)\right), \\
&\leq \exp\left(-C/2(\zeta - 1 + \log(1/\zeta))\right), &&\text{(optimal at } t^\star = 1/2(1 - 1/\zeta)) \\
&\leq \exp\left(-C/2(\zeta - 1 + \log(1/M))\right).
\end{aligned}
$$

By union bound,

$$\mathsf{P}\left(\exists k \in \{1, \ldots, \mathfrak{n}\} : \rho_k \geq \zeta C\right) \leq \mathfrak{n}\exp\left(-C/2(\zeta - 1 + \log(1/M))\right). \tag{10}$$

Let $\delta$ equals the right hand side, and solve for $\zeta$, we get for any $\delta \geq \delta_0 > 0$, with probability at least $1 - \delta$

$$\forall k : \rho_k < 2\log\left(\frac{\mathfrak{n}}{\delta}\right) + C(1 + \log M) \tag{11}$$

where $\delta_0 \geq 2\mathfrak{n}/(C(M - \log M - 1))$ is chosen to satisfy $\zeta \leq M$. We can also choose $\delta_0$ first and set $M$ accordingly.

Putting everyting together, with probability at least $1 - \delta$, the mean squared error (averaged over all output pixels)

$$
\begin{aligned}
\text{Error}^2 &= \frac{1}{|\Xi|C}\sum_{ij\in\Xi}\text{Error}_{ij}^2, \\
&\leq \left(\left(\sqrt{\mathfrak{n}\sigma^2\left(2\log\left(\frac{\mathfrak{n}}{\delta}\right) + C(1 + \log M)\right)} + \sqrt{\mathfrak{n}C}\right)B\sqrt{K_H K_W C}\right)^2 \Big/ C, \\
&= \mathcal{O}\left(\frac{K_H K_W C^2 \mathfrak{n}(1 + 1/C\log(\mathfrak{n}/\delta))}{C}\right). \tag{12}
\end{aligned}
$$

Note $K_H K_W C^2$ is the number of parameters in the convolution net.

CASE 2 — underparameterization: $|\Xi| > K_H \times K_W \times C \times C$. Using the same notation above, assuming no redundant features, we apply Lemma 2 to get the prediction of the $(i, j)$-th (multi-channel) pixel of a test example $x$ as

$$
\begin{aligned}
\hat{W}^\top \mathsf{P}_{ij}(x) &= Y^\top X(X^\top X)^{-1}\mathsf{P}_{ij}(x) \\
&= \Lambda^\top X^\top X(X^\top X)^{-1}\mathsf{P}_{ij}(x) \\
&= \Lambda^\top \mathsf{P}_{ij}(x)
\end{aligned}
$$

Recall the definition of $\Lambda$, which maps a patch to the corresponding (multi-channel) pixel. In this case, the prediction is exact, and the error is zero. Since in this case $\mathfrak{n}$, this case can be merged with equation 12. $\qquad\square$

*Proof of Lemma 1.* The proof is an extension of Theorem 1 to the case of more than one training examples. Using the notation in the Lemma, the training objective can be written as the matrix form

$$\hat{R} = \frac{1}{2}\left\| XW - Y \right\|_2^2,$$

and the gradient as

$$\frac{\partial \hat{R}}{\partial W} = X^\top (XW - Y).$$

Since the model is overparameterized, and there is no unique minimizer to the empirical risk. However, gradient descent converges to a unique solution. Note the gradient descent step is:

$$
\begin{aligned}
W^t &= W^{t-1} - \eta_t \left. \frac{\partial \hat{R}}{\partial W} \right|_{W = W_{t-1}} \\
&= W^{t-1} - \eta_t X^\top (XW^{t-1} - Y) \\
&= W^0 - \sum_{\tau=1}^{t-1} \eta_\tau X^\top (XW^\tau - Y) \\
&:= W^0 + X^\top U^t,
\end{aligned}
$$

where $U^t = -\sum_{\tau=1}^{t-1} \eta_\tau (XW^\tau - Y) \in \mathbb{R}^{N \times d}$ parameterizes the solution at iteration $t$. Since $XX^\top$ is invertible in this case. A unique solution exists under this parameterization, which can be obtained by solving

$$
\begin{aligned}
X(W^0 + X^\top \hat{U}) &= Y \\
\Rightarrow \quad \hat{U} &= \left( XX^\top \right)^{-1} (Y - XW^0).
\end{aligned}
$$

Plug this into the parameterization, we get

$$\hat{W} = W^0 + X^\top \left( XX^\top \right)^{-1} (Y - XW^0). \tag{13}$$

$\square$

*Proof of Lemma 2.* Using the same notation as in the proof of Lemma 1, since the model is underparameterized, a unique minimizer of the empirical risk exists. Directly solving for the optimality condition $\partial \hat{R} / \partial W = 0$, we get

$$\hat{W} = \left( X^\top X \right)^{-1} X^\top Y. \tag{14}$$

$\square$

# E  FULL RESULTS OF FULLY CONNECTED MULTI-LAYER NETWORKS

In this section, we present the detailed results on fully connected networks that are omitted from Section 3.1 due to space limit.

## E.1  FULLY CONNECTED LINEAR NETWORKS

Figure 11 shows the results on multi-layer *linear* networks with various number of hidden layers and hidden units. The depth of the architecture has a stronger effect on the inductive bias than the width. For example, the network with one hidden layer of dimension 2048 has 3.2M parameters, more than the 2.5M parameters of the network with three hidden layers of dimension 784. But the latter behaves less like the convex case.

## E.2  TWO-LAYER FULLY CONNECTED RELU NETWORKS

Li & Liang (2018) offer a theoretical characterization of learning in a two-layer ReLU neural network. They show that when the data consists of well separated clusters (i.e., the cluster diameters are much smaller than the distances between each cluster pair), training an overparameterized two-layer ReLU network will generalize well. To simplify the analysis, they study a special case where the weights in the top layer are randomly initialized and fixed; only the bottom layer weights are learned.

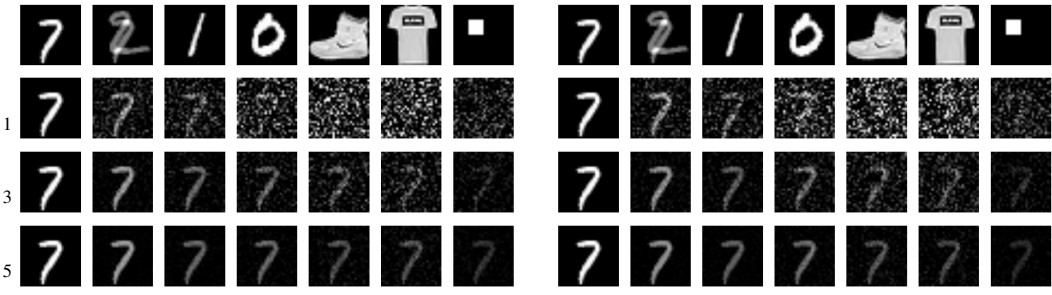

(a) Hidden dimension 784 (= input dimension)  (b) Hidden dimension 2048

Figure 11: **Visualization of predictions from trained multi-layer linear networks.** The first row shows the input images for evaluation, including the single training image "7" at the beginning of the row. The remaining rows shows the prediction from a trained linear network with 1, 3, and 5 hidden layers, respectively.

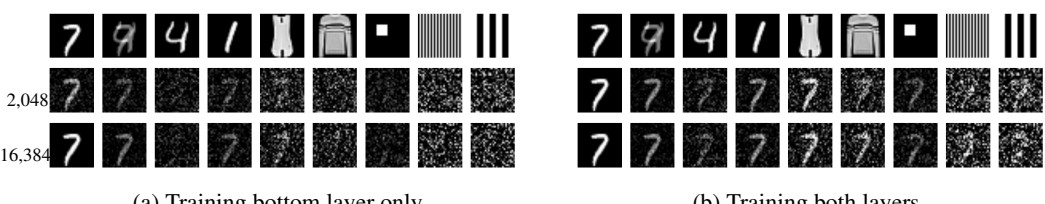

(a) Training bottom layer only  (b) Training both layers

Figure 12: **Visualization of predictions from two-layer ReLU networks.** The first row shows the input images for evaluation, including the single training image "7" at the beginning of the row. The remaining rows shows the predictions from trained models with hidden dimension 2,048 and 16,384, repetively.

We study the problem of learning a two-layer ReLU network under our identity-mapping task. Figure 12 compares the cases of learning the bottom layer only and learning both layers (left and right panels, respectively). The two cases demonstrate different inductive biases for predictions on unseen test images. When only the first layer is trained, the tendency is toward speckled noise, but when both layers are trained, the tendency is toward a constant output (the image used for training). Our observation does not contradict the theoretical results of Li & Liang (2018), which assume a well separated and clustered data distribution.

For bottom-layer only training, we can analyze it in a similar way to the one-layer networks. Let us denote

$$f_W(x) = \langle \alpha, \mathrm{ReLU}(z) \rangle, \quad z = Wx \tag{15}$$

where $W \in \mathbb{R}^{m \times d}$ is the learnable weight matrix, and $\alpha \in \mathbb{R}^m$ is randomly initialized and fixed. Although the trained weights no longer have a closed-form solution, the solution found by gradient descent is always parameterized as

$$W^T = W^0 + u^T \hat{x}^\top \tag{16}$$

where $\hat{x}$ is the training example, and $u^T \in \mathbb{R}^m$ summarizes the efforts of gradient descent up to time $T$. In particular, the gradient of the empirical risk $\hat{R}$ with respect to each row $W_{:r}$ of the learnable weight is

$$\frac{\partial \hat{R}}{\partial W_{:r}} = \frac{\partial \hat{R}}{\partial z_r} \frac{\partial z_r}{\partial W_{:r}} = \frac{\partial \hat{R}}{\partial z_r} \hat{x}^\top \tag{17}$$

Putting it together, the full gradient is

$$\frac{\partial \hat{R}}{\partial W} = \frac{\partial \hat{R}}{\partial z} \cdot \hat{x}^\top \tag{18}$$

Since the gradient lives in the span of the training example $\hat{x}$, the solution found by gradient descent is always parameterized as equation 16.

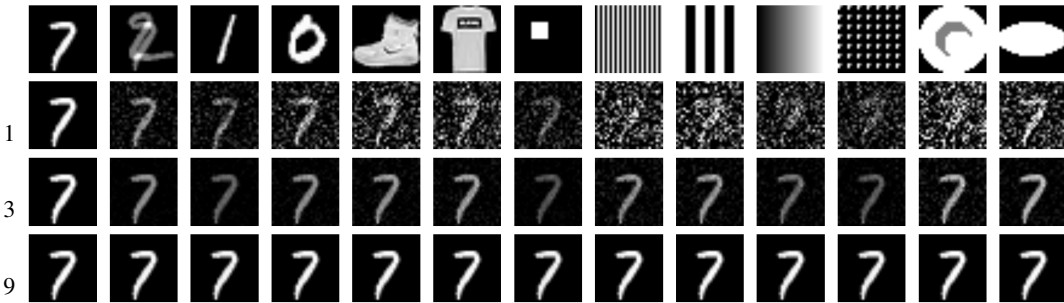

Figure 13: **Visualization of predictions from multi-layer ReLU networks.** The first row shows the input images for evaluation, including the single training image "7" at the beginning of the row. The remaining rows show the predictions from trained multi-layer ReLU FCNs with 1, 3, and 9 hidden layers.

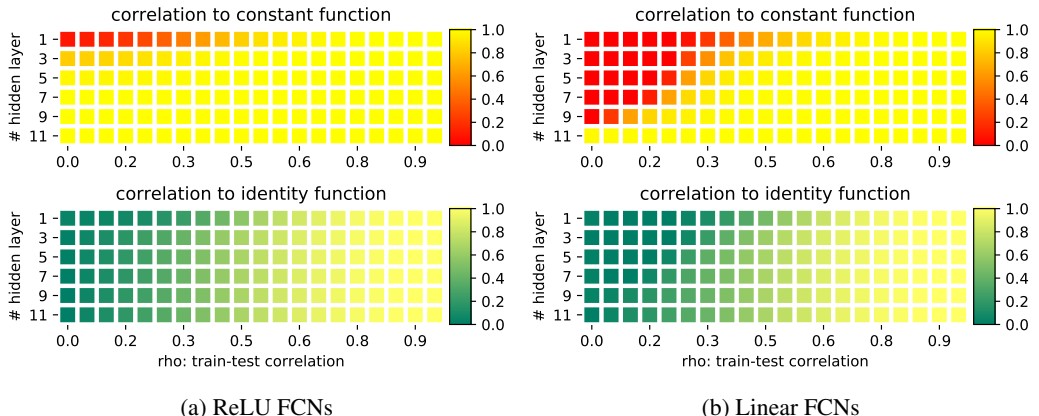

(a) ReLU FCNs                    (b) Linear FCNs

Figure 14: **Quantitative evaluation of the learned model on randomly generated test samples at various angles (correlation) to the training image.** The horizontal axis shows the train-test correlation, while the vertical axis indicate the number of hidden layers for the FCNs being evaluated. The heatmap shows the similarity (measured in correlation) between the model prediction and the reference function (the constant function or the identity function). (a) shows the results for FCNs with the ReLU activation function; (b) shows the results for linear FCNs.

The same arguments applies to multi-layer neural networks. The prediction on any test example that is orthogonal to $\hat{x}$ will depend only on randomly initialized $W^0$ and upper layer weights. When only the bottom layer is trained, the upper layer weights will also be independent from the data, therefore the prediction is completely random. However, when all the layers are jointly trained, the arguments no longer apply. The empirical results presented in the main text that multi-layer networks bias towards the constant function verify this.

In particular, if the test example is orthogonal to $\hat{x}$ (i.e., $\hat{x}^\top x = 0$), the prediction depends solely on the randomly initialized values in $W^0$ and therefore can be characterized by the distribution used for parameter initialization.

However, when both layers are trained, the upper layer weights are also tuned to make the prediction fit the training output. In particular, the learned weights in the upper layer depend on $W^0$. Therefore, the randomness arguments shown above no longer apply even for test examples orthogonal to $\hat{x}$. As the empirical results show, the behavior is indeed different.

### E.3 NONLINEAR MULTI-LAYER FULLY CONNECTED NETWORKS

In this section, we consider the general case of multilayer fully connected networks (FCNs) with ReLU activation functions. Figure 13 visualizes predictions from trained ReLU FCNs with 1, 3, or 9

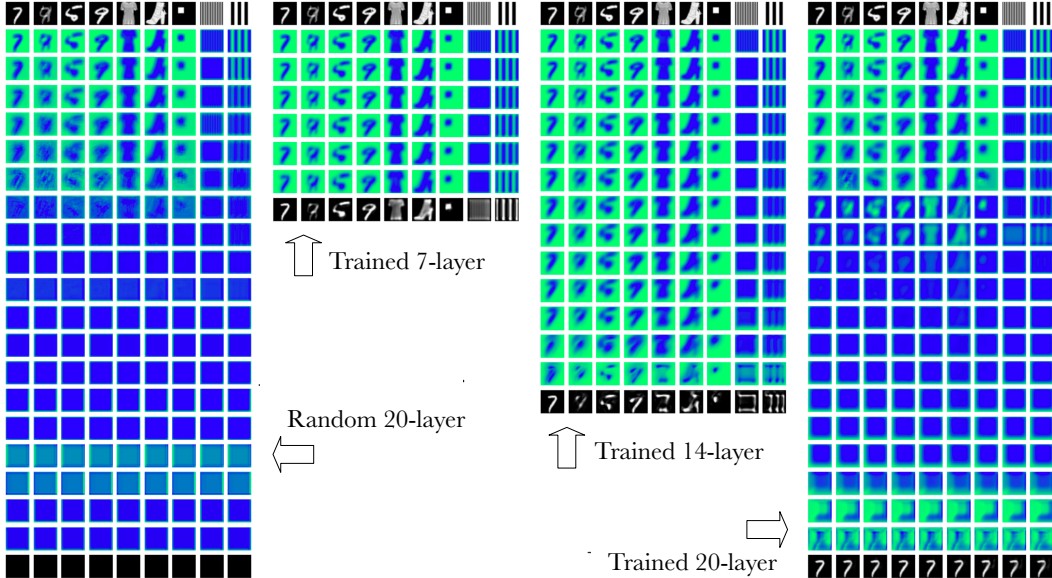

Figure 15: **Visualization the intermediate layers of CNNs with different number of layers.** The first column shows a randomly initialized 20-layer CNN (random shallower CNNs look similar to the truncation of this). The rest of the columns show the trained CNNs with various number of layers.

hidden layers. The deepest network encodes the constant map with high confidence; the shallowest network shows behavior similar to that of a one-layer linear net. Quantitative evaluations of the behaviors are shown in Figure 14, computed in the same way as Figure 4 for CNNs (see Section 3.2). The results for linear FCNs are also shown for comparison. The linear and ReLU FCNs behave similarly when measuring the correlation to the identity function: neither of them performs well for test images that are nearly orthogonal to $\hat{x}$. For the correlation to the constant function, ReLU FCNs overfit sooner than linear FCNs when the depth increases. This is consistent with our previous visual inspections: for shallow models, the networks learn neither the constant nor the identity function, as the predictions on nearly orthogonal examples are random.

## F  VISUALIZATION OF THE INTERMEDIATE LAYER REPRESENTATIONS FOR CNNS

Unlike the FCNs, CNNs preserve the spatial relation between neurons in the hidden layers, so we can easily visualize the intermediate layers as images in comparison to the inputs and outputs, to gain more insights on how the networks are computing the functions layer-by-layer. In Figure 15, we visualize the intermediate layer representations on some test patterns for CNNs with different depths. In particular, for each example, the outputs from a convolutional layer in an intermediate layer is a three dimensional tensor of shape (#channel, height, width). To get a compact visualization for multiple channels in each layer, we compute flatten the 3D tensor to a matrix of shape (#channel, height × width), compute SVD and visualize the top singular vector as a one-channel image.

In the first column, we visualize a 20-layer CNN at random initialization[3]. As expected, the randomly initialized convolutional layers gradually smooth out the input images. The shape of the input images are (visually) wiped out after around 8 layers of (random) convolution. On the right of the figure, we show several trained CNNs with increasing depths. For a 7-layer CNN, the holistic structure of inputs are still visible all the way to the top at random initialization. After training, the network approximately renders an identity function at the output, and the intermediate activations also become less blurry. Next we show a 14-layer CNN, which fails to learn the identity function.

---

[3]Shallower CNNs at random initialization can be well represented by looking at a (top) subset of the visualization.

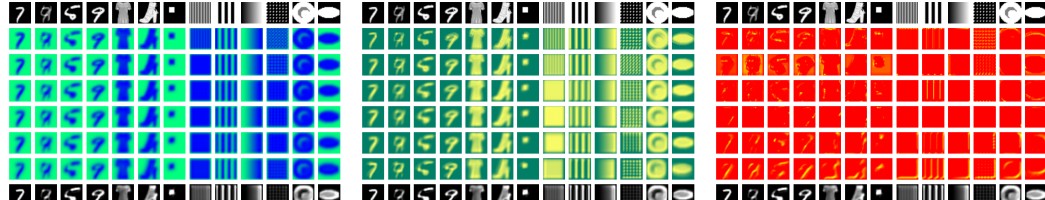

Figure 16: **Visualizing the intermediate layers of a trained 7-layer CNN.** The three subfigures show for each layer: 1) the top singular vector across the channels; 2) the channel that maximally correlate with the input image; 2) a random channel, respectively.

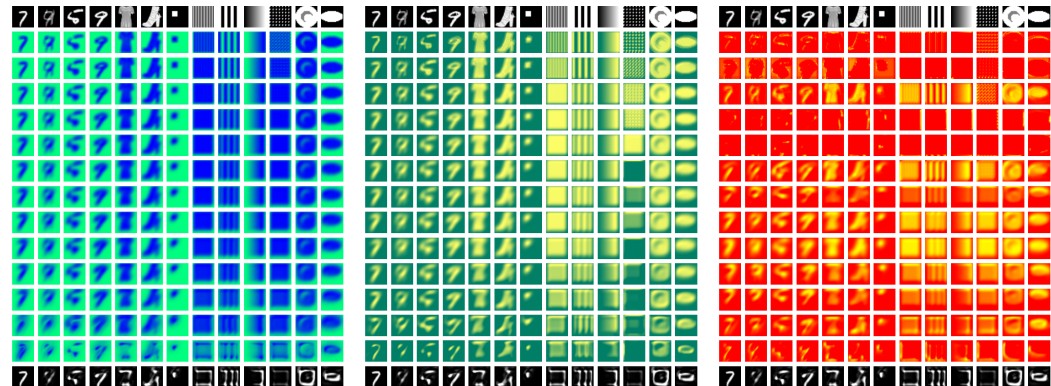

Figure 17: **Visualizing the intermediate layers of a trained 14-layer CNN.** The three subfigures show for each layer: 1) the top singular vector across the channels; 2) the channel that maximally correlate with the input image; 2) a random channel, respectively.

However, it manages to recover meaningful information in the higher layer activations that were (visually) lost in the random initialization. On the other hand, in the last column, the network is so deep that it fails to make connection from the input to the output. Instead, the network start from scratch and constructs the digit '7' from empty and predict everything as '7'. However, note that around layer-8, we see the activations depict slightly more clear structures than the randomly initialized network. This suggests that some efforts have been made during the learning, as opposed to the case that the bottom layers not being learned due to complete gradient vanishing. Please refer to Appendix G for further details related to potential gradient vanishing problems.

Two alternative visualizations to the intermediate multi-channel representations are provided that show the channel that is maximally correlated with the input image, and a random channel (channel 0). Figure 16, Figure 17 and Figure 18 illustrate a 7-layer CNN, a 14-layer CNN and a 20-layer CNN, respectively.

## G  MEASURING THE CHANGE IN WEIGHTS OF LAYERS POST TRAINING

In this section, we study the connection between the inductive bias of learning the constant function and the potential gradient vanishing problem. Instead of measuring the norm of gradient during training, we use a simple proxy that directly compute the distance of the weight tensor before and after training. In particular, for each weight tensor $W^0$ at initialization and $W^\star$ after training, we compute the relative $\ell_2$ distance as

$$d(W^0, W^\star) \coloneqq \frac{\|W^0 - W^\star\|}{\|W^0\|}$$

The results for CNNs with various depths are plotted in Figure 19(a). As a general pattern, we do see that as the network architecture gets deeper, the distances at lower layers do become smaller. But they are still non-zero, which is consistent with the visualization in Figure 15 showing that even

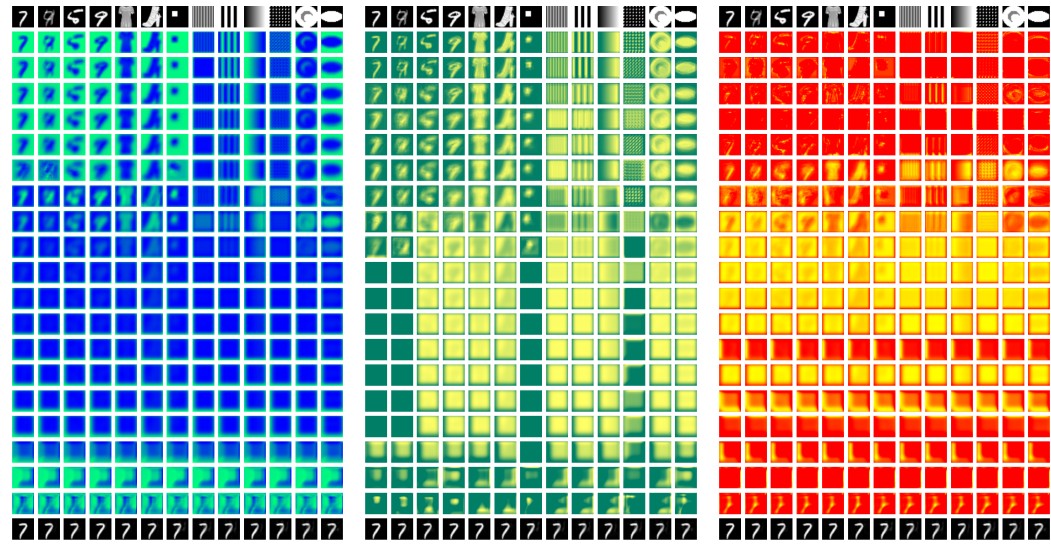

Figure 18: **Visualizing the intermediate layers of a trained 20-layer CNN.** The three subfigures show for each layer: 1) the top singular vector across the channels; 2) the channel that maximally correlate with the input image; 2) a random channel, respectively.

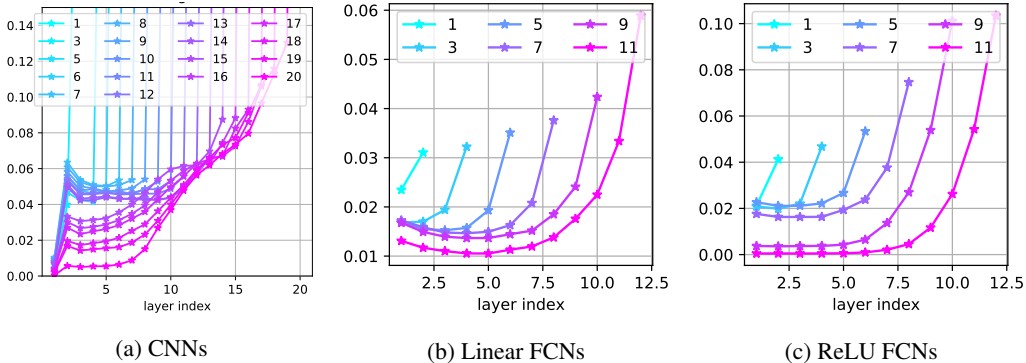

(a) CNNs                (b) Linear FCNs                (c) ReLU FCNs

Figure 19: **The relative $\ell_2$ distance of the weight tensors before and after training at each layer.** The curves compare models at different depth. Most of the networks have significantly larger distances on the top-most layer. To see a better resolution at the bottom layers, we cut off the top layer in the figures by manually restricting the y axis.

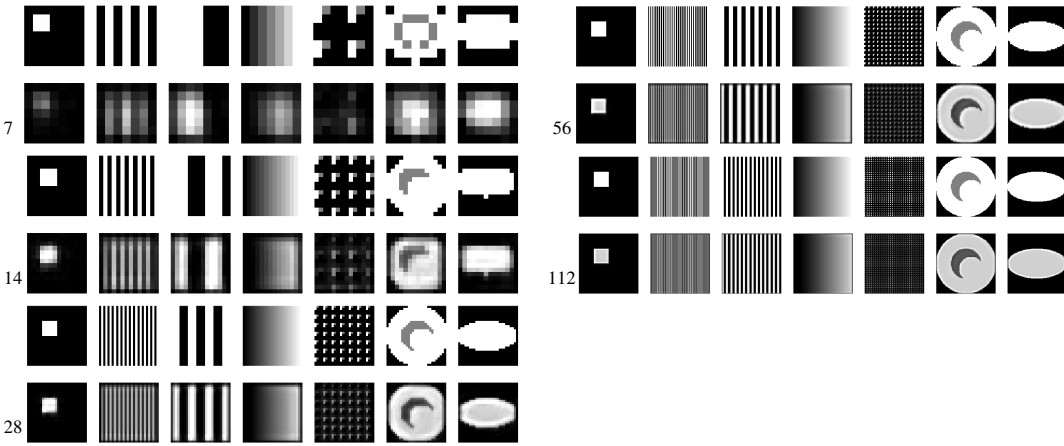

Figure 20: **Visualization of a 5-layer CNN on test images of different sizes.** Every two rows show the inputs and model predictions. The numbers on the left indicate the input image size (both width and height).

for the 20-layer CNN, where the output layer fits to the constant function, the lower layers does get enough updates to allow them to be visually distinguished from the random initialization.

In Figure 19(b) and (c), we show the same plots for linear FCNs and FCNs with ReLU activation, respectively. We see that especially for ReLU FCN with 11 hidden layers, the distances for the weight tensors at the lower 5 layers are near zero. However, recall from Figure 13 in Section E.3, the ReLU FCNs start to bias towards the constant function with only three hidden layers, which are by no means suffering from vanishing gradients as the plots here demonstrate.

## H    FULL RESULTS ON ROBUSTNESS OF INDUCTIVE BIASES

### H.1    TESTING ON DIFFERENT INPUT IMAGE SIZES

Figure 6 in Section 3.3 visualize the predictions of a 5-layer CNN on inputs of the extreme sizes of $7 \times 7$ and $112 \times 112$. Here in Figure 20 we provide more results on some intermediate image sizes.

We also test a trained 20-layer CNN on various input sizes (Figure 7). It is interesting that the constant map also holds robustly across a wide range of input sizes. Especially from the prediction on large input images, we found hints on how the CNNs actually compute the constant function. It seems the artifacts from the outside boundary due to convolution padding are used as cues to "grow" a pattern inwards to generate the digit "7". The visualizations of the intermediate layer representatiosn in Figure 21 is consistent with our hypothesis: the CNN take the last several layers to realize this construction. This is very clever, because in CNNs, the same filter and bias is applied to all the spatial locations. Without relying on the artifacts on the boundaries, it would be very challenging to get a sense of the spatial location in order to construct an image with a holistic structure (e.g. the digit "7" in our case).

### H.2    THE UPPER SUBNETWORKS

In the visualization of intermediate layers (Figure 15), the intermediate layers actually represent the "lower" subnetwork from the inputs. Here we investigate the "upper" subnetwork. Thanks again to the spatial structure of CNNs, we can skip the lower layers and feed the test patterns directly to the intermediate layers and still get interpretable visualizations[4]. Figure 22 shows the results for the top-one layer from CNNs with various depths. A clear distinction can be found at 15-layer CNN,

---

[4]Specifically, the intermediate layers expect inputs with multiple channels, so we repeat the grayscale inputs across channels to match the expected input shape.

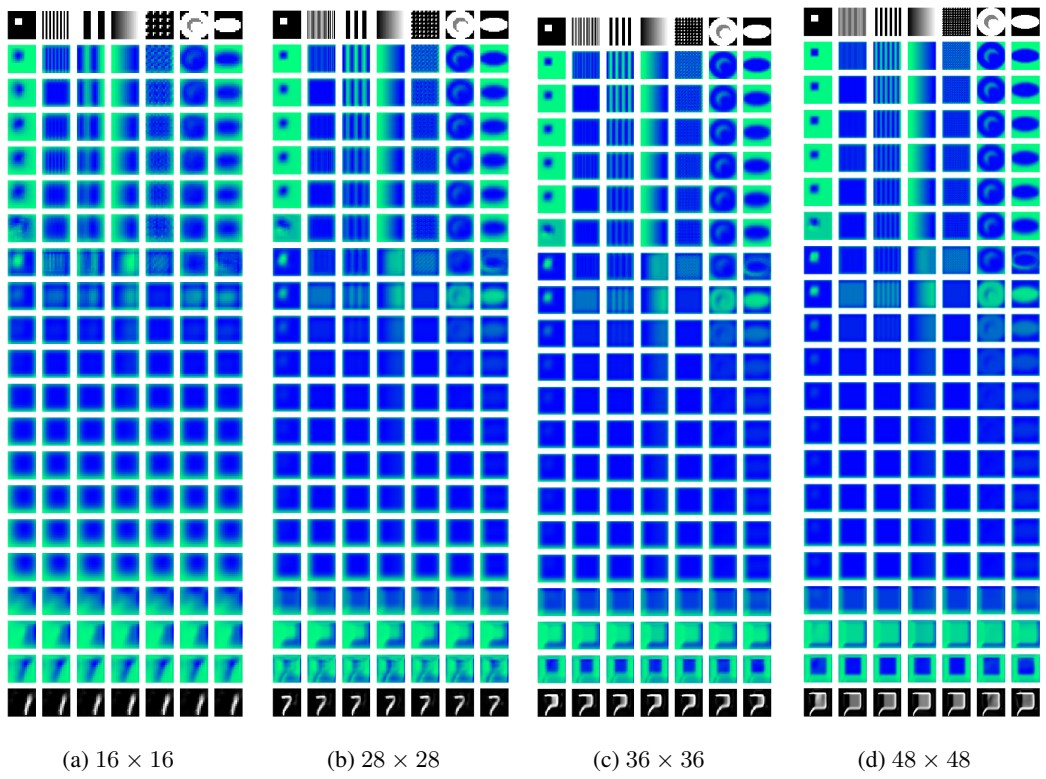

(a) 16 × 16    (b) 28 × 28    (c) 36 × 36    (d) 48 × 48

Figure 21: **Visualization of intermediate representations when testing on images of different sizes from training images for a 20-layer trained CNN**. The CNN is trained on a 28 × 28 image of the digit "7".

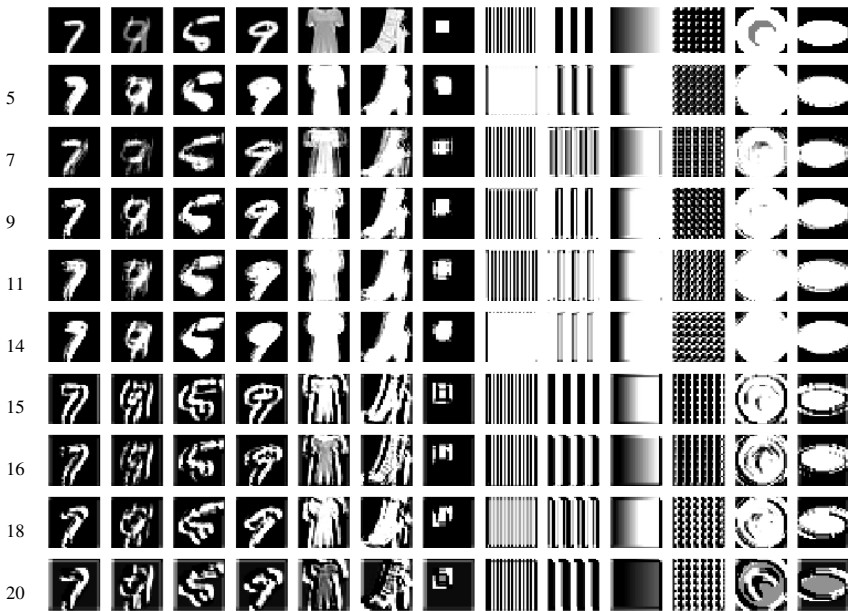

Figure 22: **Visualizing only the final layer in trained networks.** The first row are the input images, which are directly fed into the final layer of trained networks (skipping the bottom layers). The remaining rows shows the predictions from the top layers of CNNs, with the numbers on the left indicating their (original) depth.

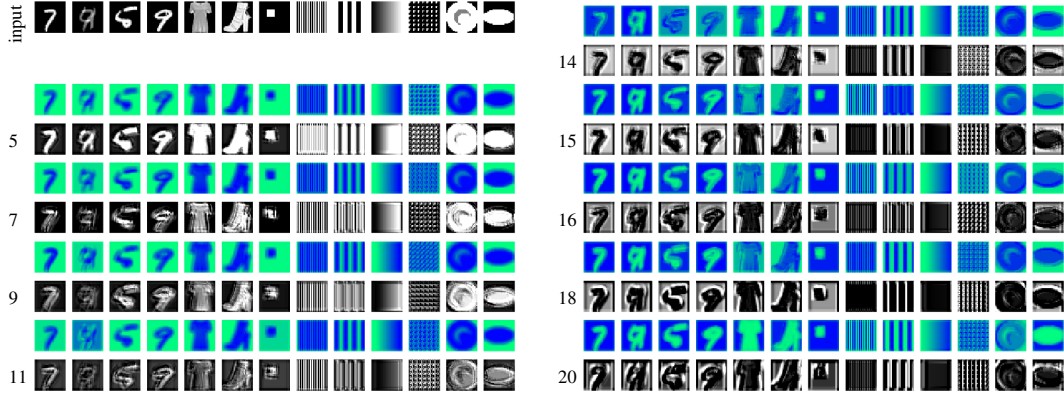

Figure 23: **Visualizing only the top two layers in trained networks.** The first row are the input images, which are directly fed into the top two layer of trained networks (skipping the bottom layers). The remaining rows shows the predictions from the top two layers of CNNs, with the numbers on the left indicating their (original) depth. More specifically, each of the two top layers occupies one row. The colorful rows are the visualizations (as the top singular vector across channels) of the outputs of the second to the last layer from each network. The grayscale rows are the outputs of the final layer from each network.

which according to Figure 3 is where the networks start to bias away from edge detector and towards the constant function.

The predictions from the final two layers of each network are visualized in Figure 23. Figure 24 focuses on the 20-layer CNN that learns the constant map, and visualize the upper 3 layers, 6 layers and 10 layers, respectively. In particular, the last visualization shows that the 20-layer CNN is already starting to construct the digit "7" from nowhere when using only the upper half of the model.

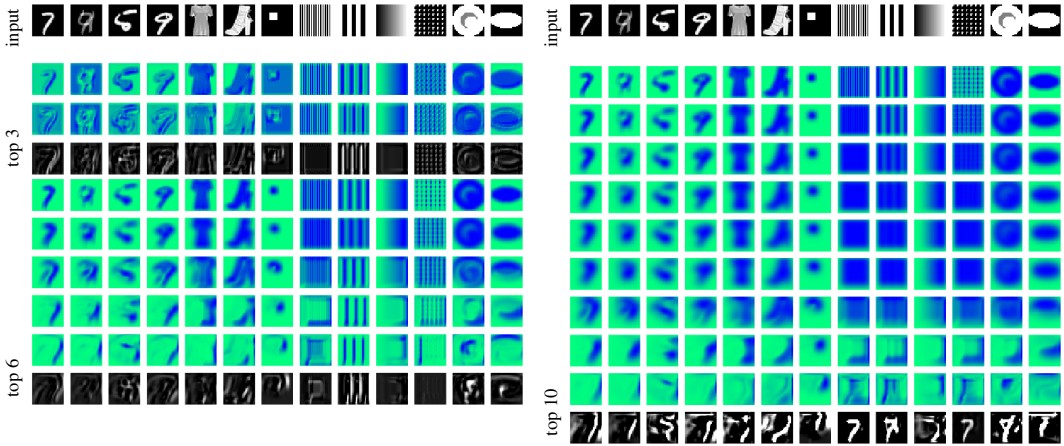

Figure 24: **Visualzing the top 3 layers, 6 layers and 10 layers of a 20-layer CNN.** Visualizations formatted in the same way as Figure 23.

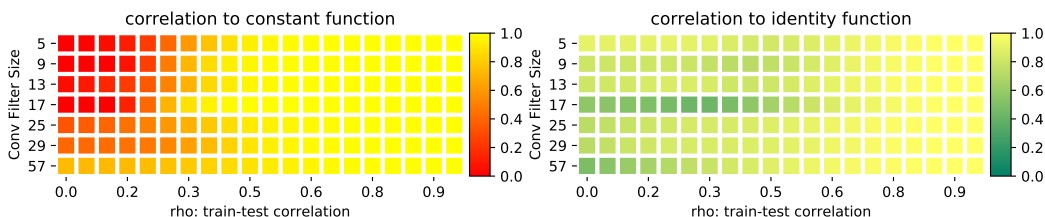

Figure 25: **Inductive bias of a 5-layer CNN with varying convolutional filter size.** The heatmap is arranged similarly as Figure 4, except that the rows correspond to CNNs filter sizes.

## I  FULL RESULTS ON VARYING DIFFERENT FACTORS WHEN TRAINING CNNS

The study on the effects of various hyperparameters on the inductive bias of trained CNNs is briefly presented in Section 3.4. The full results are shown here.

**Convolution filter sizes**  Figure 25 and Figure 26 illustrate the inductive bias with varying convolution filter size from $5 \times 5$ to $57 \times 57$. The visualization shows that the predictions become more and more blurry as the filter sizes grow. The heatmaps, especially the correlation to the identity function, are not as helpful in this case as the correlation metric is not very good at distinguishing images with different levels of blurry. With extremely large filter sizes that cover the whole inputs, the CNNs start to bias towards the constant function. Note our training inputs are of size $28 \times 28$, so $29 \times 29$ filter size allows all the neurons to see no less than half of the spatial domain from the previous layer. $57 \times 57$ filters centered at any location within the image will be able to see the whole previous layer. On the other hand, the repeated application of *the same* convolution filter through out the spatial domain is still used (with very large boundary paddings in the inputs). So the CNNs are *not* trivially doing the same computation as FCNs.

**Convolution channel depths**  Figure 10 in Section 3.4 shows that the learned identity function is severely corrupted when only 3 channels are used in the 5-layer CNNs. Figure 27 presents the correlation to the constant and the identity function when different numbers of convolution channels are used. The heatmap is consistent with the visualizations, showing that the 5-layer CNN fails to approximate the identity function when only three channels are used in each convolution layer.

Furthermore, Figure 28 visualize the predictions of trained 3-channel CNNs with various depths. The 3-channel CNNs beyond 8 layers fail to converge during training. The 5-layer and the 7-layer CNNs implement functions biased towards edge-detecting or countour-finding. But the 6-layer and the 8-layer CNNs demonstrate very different biases. The potential reason is that with only a few

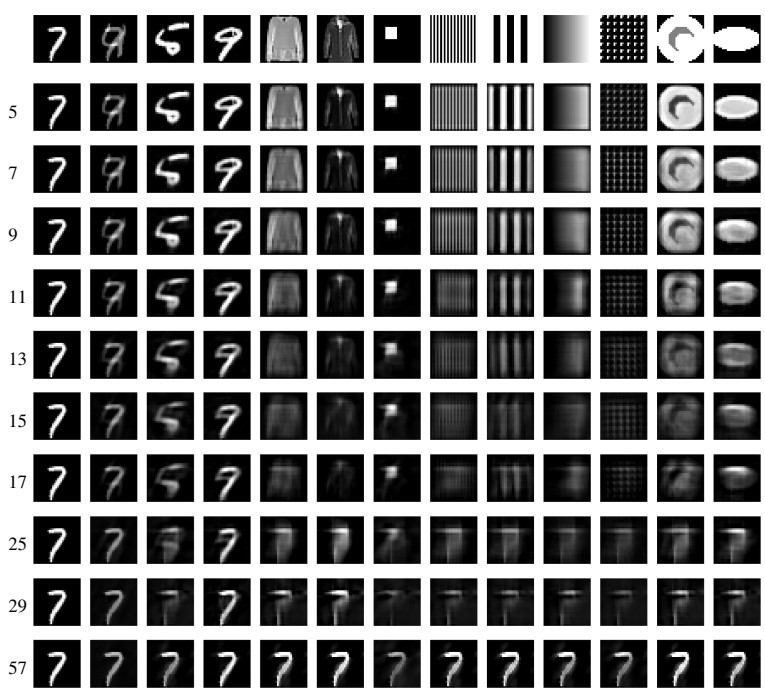

Figure 26: **Visualizing the predictions from CNNs with various filter sizes.** The first row is the inputs, including the single training image "7". The remaining rows are predictions, with the numbers on the left showing the corresponding filter sizes.

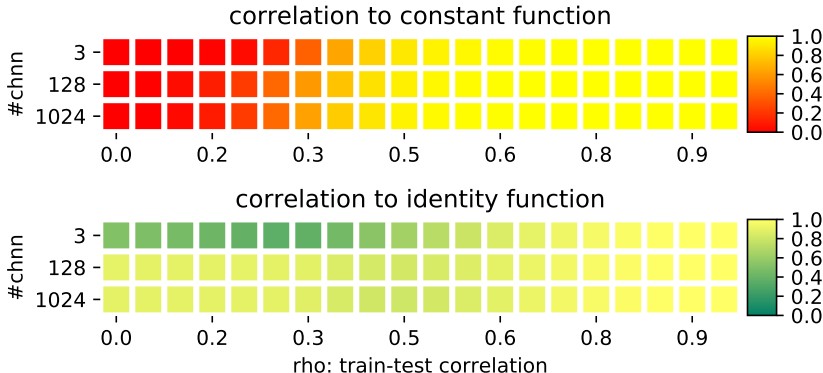

Figure 27: **Correlation to the constant and the identity function for different convolution channels in a 5-layer CNN.**

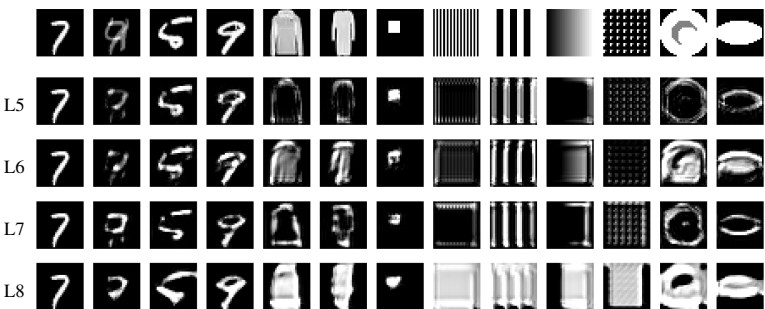

Figure 28: **Visualization predictions from CNNs with 3 convolution channels and with various number of layers (numbers on the left).** The first row is the inputs, and the remaining rows illustrate the network predictions.

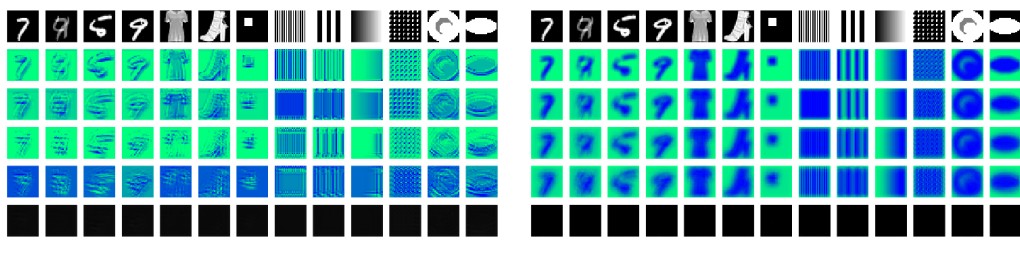

(a) 3 channels, random init      (b) 128 channels, random init

Figure 29: **Visualizing the randomly initialized models to compare two 5-layer CNNs with 3 convolution channels per layer and 128 convolution channels per layer, respectively.** The subfigures visualize the predictions of intermediate layers of the two network at random initialization. The multi-channel intermediate layers are visualized as the top singular vectors.

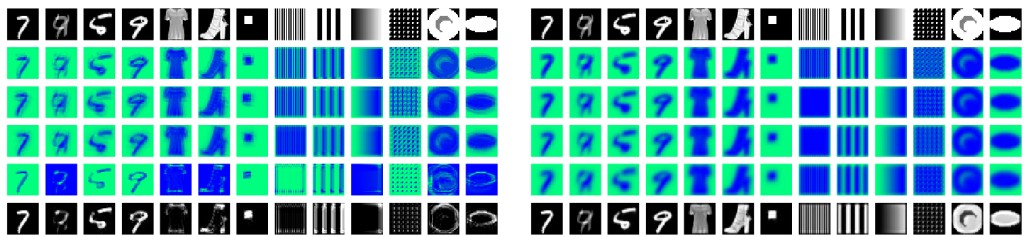

(a) 3 channels, after training      (b) 128 channels, after training

Figure 30: **Comparing two 5-layer CNNs with 3 convolution channels per layer and 128 convolution channels per layer, respectively.** Layout is similar to Figure 29.

channels, the random initialization does not have enough randomness to smooth out "unlucky" bad cases. Therefore, the networks have higher chance to converge to various corner cases. Figure 29 and Figure 30 compare the random initialization with the converged network for a 3-channel CNN and a 128-channel CNN. From the visualizations of the intermediate layers, the 128-channel CNN already behave more smoothly than the 3-channel CNN at initialization.

**Initialization schemes** In transfer learning, it is well known that initializing with pre-trained network weights could affect the inductive bias of trained models. On the other hand, different *random* initialization schemes are mainly proposed to help with optimization by maintaining information flow or norms of representations at different depths. It turns out that they also strongly affect the inductive bias of the trained models. Figure 31 visualizes the different inductive biases. Let $f_i$, $f_o$ be the *fan in* and *fan out* of the layer being initialized. We tested the following commonly used initialization schemes: 1) default: $\mathcal{N}(0, \sigma^2 = 1/(f_i f_o))$; 2) Xavier (a.k.a. Glorot) init (Glorot & Bengio, 2010): $\mathcal{N}(0, \sigma^2 = 2/(f_i + f_o))$; 3) Kaiming init (He et al., 2015): $\mathcal{N}(0, \sigma^2 = 2/f_i)$; 4) Orthogonal init (Saxe et al., 2014). Variations with uniform distributions instead of Gaussian distributions are also evaluated for Xavier and Kaiming inits. All initialization schemes bias toward the identity function for shallow networks. But Kaiming init produces heavy artifacts on test predictions. For the bias towards the constant function in deep networks, Xavier init behaves similarly to the default init scheme, though more layers are needed to learn a visually good identity function. On the other hand, the corresponding results from the Kaiming init is less interpretable.

**Optimizers** First order stochastic optimizers are dorminately used in deep learning due to the huge model sizes and dataset sizes. To improve convergence speed, various adaptive methods are introduced (Duchi et al., 2011; Kingma & Ba, 2014; Graves, 2013). It is known that those methods lead to worse generalization performances in some applications (e.g. Wu et al. (2016)). But they are extremely popular in practice due to the superior convergence speed and easier hyper-parameter tuning than the vanilla SGD. We compare several popular optimizers in our framework, and confirm that different optimizers find different global minimizers, and those minimizers show drastically

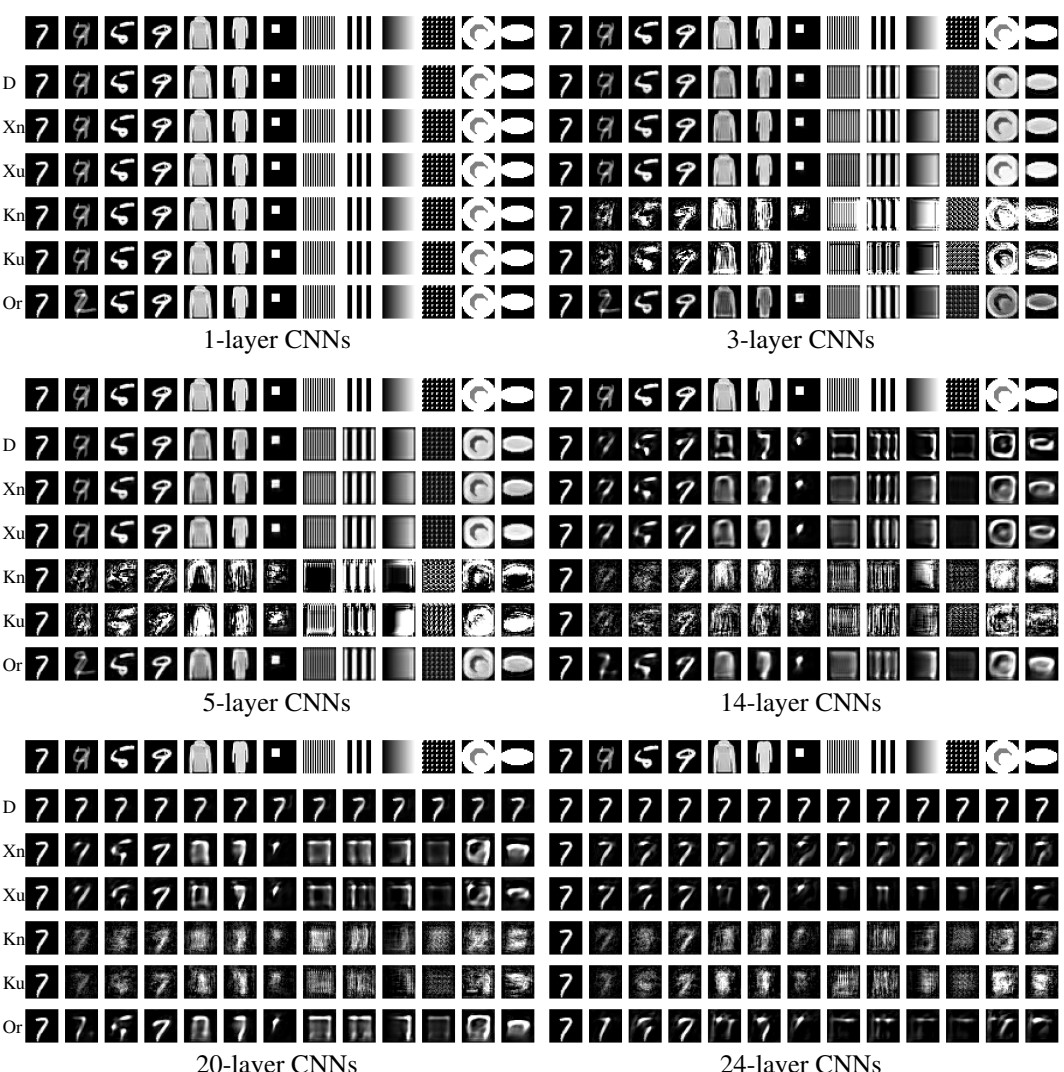

Figure 31: **Visualization of the predictions from CNNs trained with the D) default, Xn) Xavier normal, Xu) Xavier uniform, Kn) Kaiming normal, Ku) Kaiming uniform, and Or) orthogonal initialization schemes.** The first row shows the inputs, and the remaining rows shows the predictions from each trained networks.

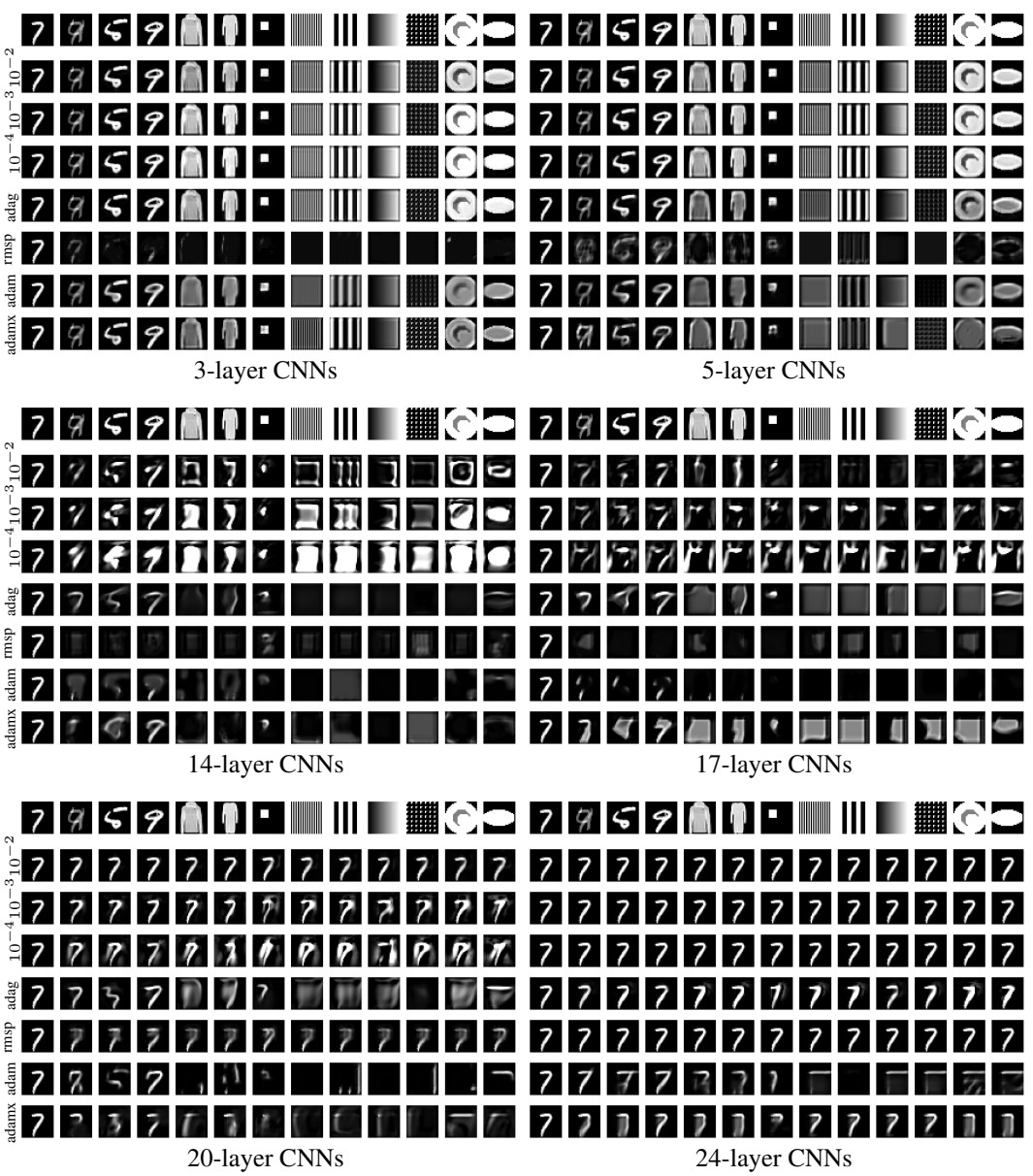

Figure 32: **Visualization of the predictions from CNNs trained with different optimizers.** The first row shows the inputs, and the remaining rows shows the predictions from SGD (lr 0.01), SGD (lr 0.001), SGD (lr 0.0001), Adagrad, RMSprop, Aam, and Adamax respectively.

different inductive biases on test examples, as shown in Figure 32. Some optimizer requires a smaller base learning rate to avoid parameter exploding during training. In particular, we use base learning rate 0.001 for Adagrad and Adamax, 0.0001 for Adam and RMSprop. For comparison, we also include results from SGD with those corresponding base learning rates.

## J CORRELATION VS MSE

Figure 33, Figure 34 and Figure 35 can be compared to their corresponding figures in the main text. The figures here are plotted with the MSE metric between the prediction and the groundtruth, while the figures in the main text uses the correlation metric. Each corresponding pair of plots are overall consistent. But the correlation plots show the patterns more clearly and has a fixed value range of [0, 1] that is easier to interpret.

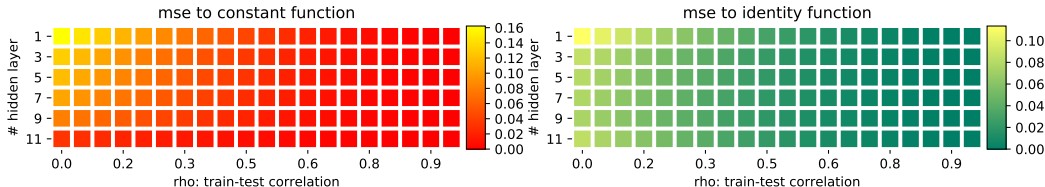

Figure 33: **Quantitative evaluation of linear FCNs.** The same as Figure 14(a), except MSE is plotted here instead of correlation.

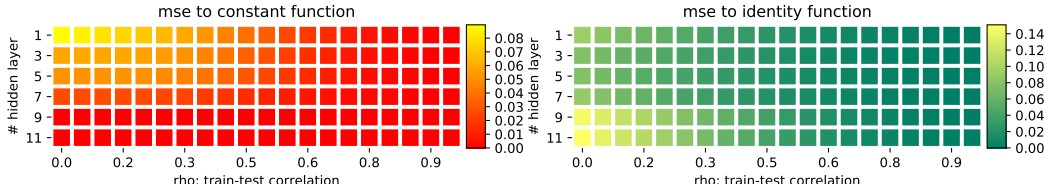

Figure 34: **Quantitative evaluation of ReLU FCNs.** The same as Figure 14(b), except MSE is plotted here instead of correlation.

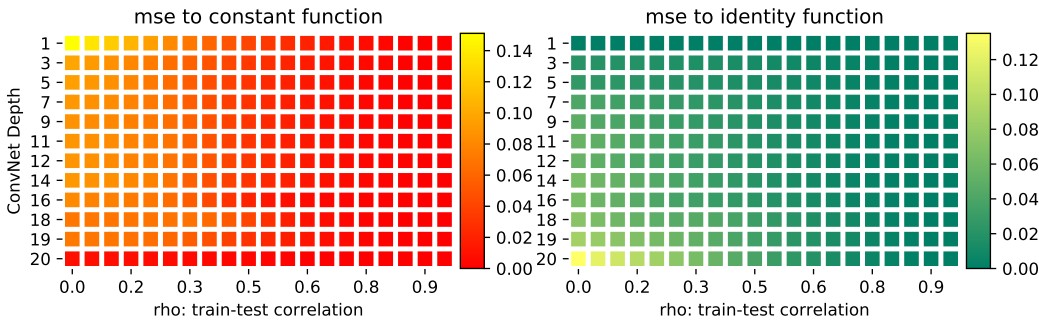

Figure 35: **Quantitative evaluation of CNNs.** The same as Figure 4, except MSE is plotted here instead of correlation.

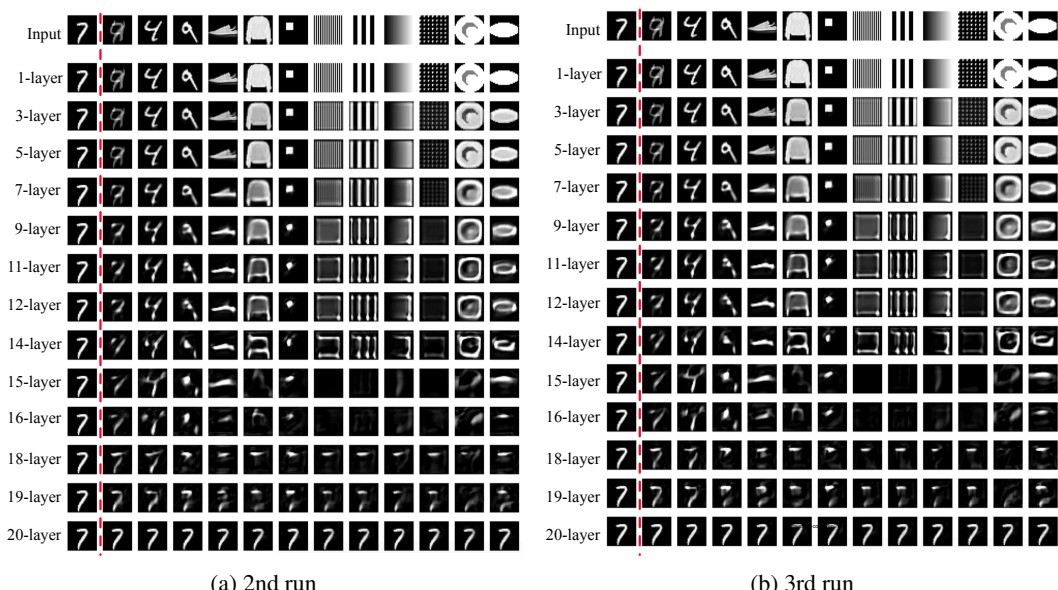

(a) 2nd run

(b) 3rd run

Figure 36: **Visualization of predictions from CNNs trained on a single example.** The same training example is used as in the main text, but two extra runs of training and evaluation are listed to show the robustness of our main observations to the randomness in the experiments.

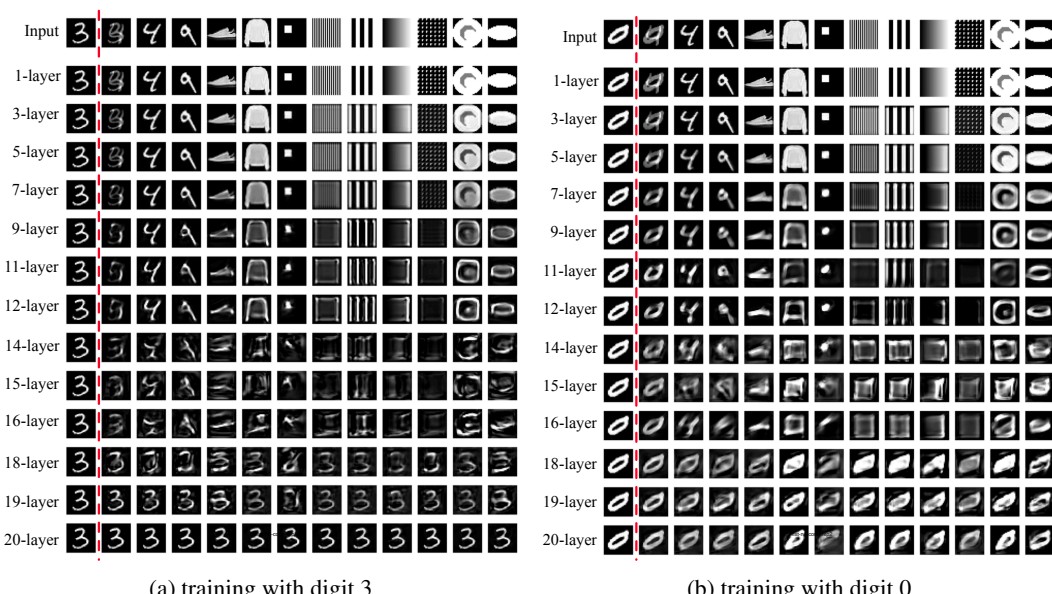

(a) training with digit 3

(b) training with digit 0

Figure 37: **Visualization of predictions from CNNs trained on a single example.** Two different (randomly chosen) training images (a) digit 3, (b) digit 0, are shown to compare robustness of our main observations to different training images.

## K    ROBUSTNESS OF OBSERVATIONS TO RANDOM SEEDS

We evaluate the robustness of our main observations to randomness from the experiments in this section. In Figure 36, two different runs of training and evaluation are compared side by side, and the results are very consistent. In Figure 37 we further perturb the random seeds for data loading, so that different (single) training images are loaded for training. We can see that the main observations

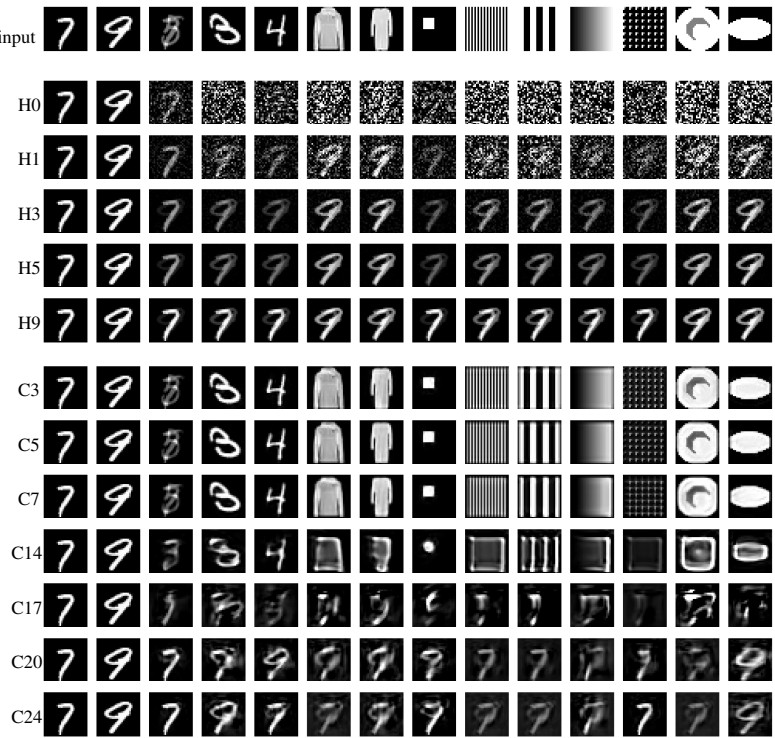

Figure 38: **Visualization of the predictions from networks trained with 2 examples**. The first two columns are training examples, and the remaining columns are unseen test cases. H0—H9 shows fully connected networks with the corresponding number of hidden layers. C3—C24 shows CNNs with the corresponding number of layers.

hold for both cases: CNNs learn the identity map, edge detector and the constant map as the depth increases.

## L  LEARNING WITH MULTIPLE EXAMPLES

We focus on learning from a single example in this paper because the two extreme inductive biases of the *identity map* and the *constant map* can be precisely defined and tested against. When training with multiple examples, the constant map is no longer a viable solution to the optimization problem. Nevertheless, we can still qualitatively evaluate the inductive bias via visualization of predictions. Figure 38 shows the results on various networks trained with two examples. In particular, for networks that are known to overfit to the constant map when trained with one example (e.g. fully connected networks with 9 hidden layers, and 20-layer convolutional networks), similar overfitting behaviors are observed. In this case, a proper definition of a *constant map* no longer holds, as the network perfectly reconstruct each individual digit from the training set. On the test set, a notion of memorization can be recognized as always predicting a mixture of the training samples. Note sometimes the prediction is biased more towards one of the training samples depending on the input patterns.

Figure 39 shows the results with three examples, with similar observations. Figure 40 shows the situation when training with the full MNIST training set (60k examples). In this case, even the deepest convolutional networks we tried successfully learn the identity function. Fully connected networks learn the identity function on the manifold of digit inputs, but still cannot reconstruct meaningful results on other test patterns.

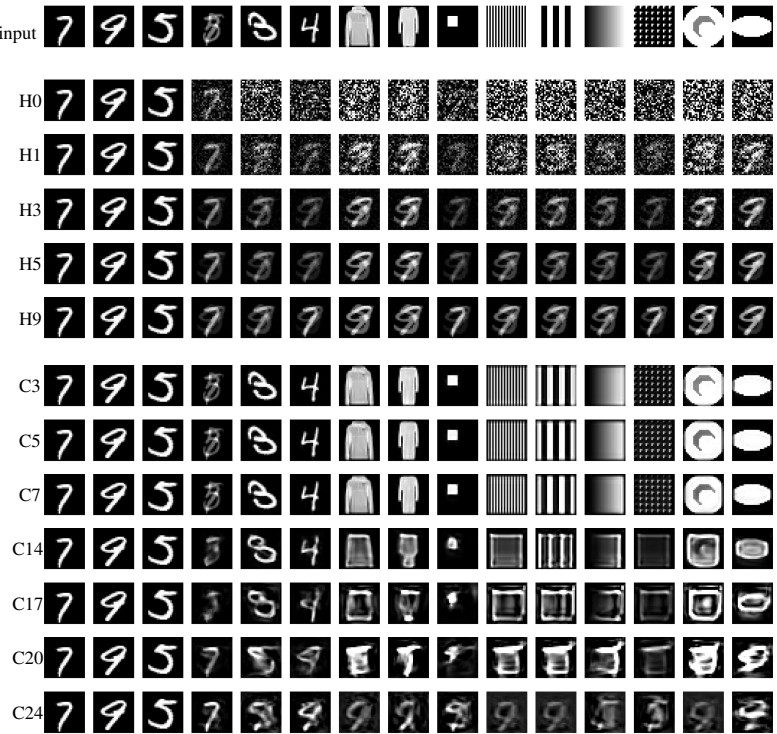

Figure 39: **Visualization of the predictions from networks trained with 3 examples**. The first three columns are training examples, and the remaining columns are unseen test cases. H0—H9 shows fully connected networks with the corresponding number of hidden layers. C3—C24 shows CNNs with the corresponding number of layers.

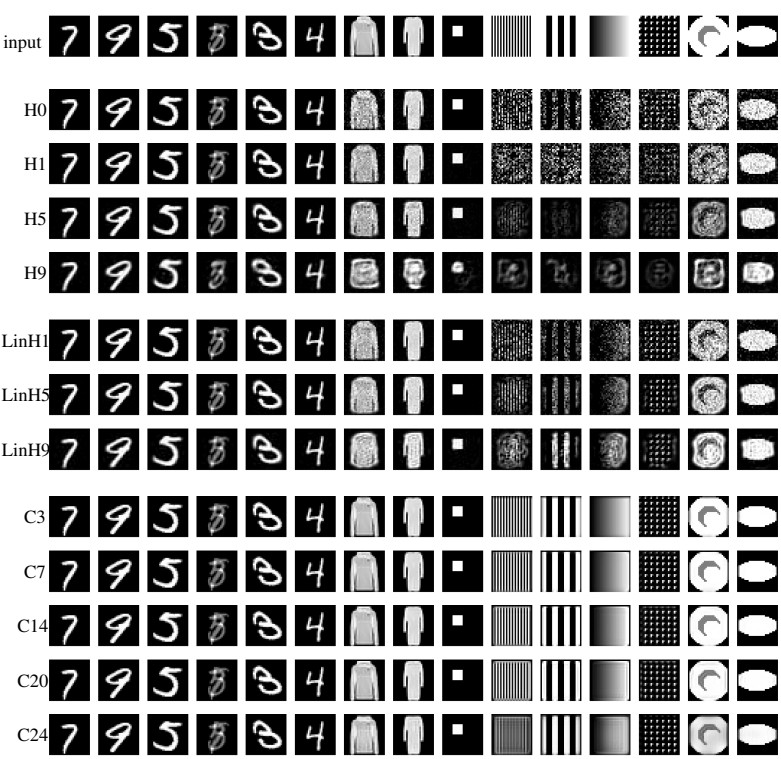

Figure 40: **Visualization of the predictions from networks trained with 60k examples**. The first three columns are (3 out of 60k) training examples, and the remaining columns are unseen test cases. H0—H9 show fully connected networks (ReLU activation) with the corresponding number of hidden layers. LinH1—LinH9 show linear fully connected networks (without activation) with the corresponding number of hidden layers. And C3—C24 show CNNs with the corresponding number of layers.

| FCN num layers | 1 | 2 | 4 | 6 | 8 | 10 |
|---|---|---|---|---|---|---|
| hidden dim = 784 | 614,656 | 1,229,312 | 2,458,624 | 3,687,936 | 4,917,248 | 6,146,560 |
| hidden dim = 2048 | | 3,211,264 | 11,599,872 | 19,988,480 | 28,377,088 | 36,765,696 |

| CNN num layers | 1 | 3 | 5 | 6 | 7 | 8 |
|---|---|---|---|---|---|---|
| $5 \times 5$ filter, 3 channels | 25 | 375 | 825 | 1,050 | 1,275 | 1,500 |
| $5 \times 5$ filter, 128 channels | 25 | 416,000 | 1,235,200 | 1,644,800 | 2,054,400 | 2,464,000 |
| $5 \times 5$ filter, 1024 channels | | | 78,694,400 | | 131,123,200 | |
| $7 \times 7$ filter, 3 channels | 49 | 735 | 1,617 | 2,058 | 2,499 | 2,940 |
| $7 \times 7$ filter, 128 channels | 49 | 815,360 | 2,420,992 | 3,223,808 | 4,026,624 | 4,829,440 |
| $7 \times 7$ filter, 1024 channels | | | 154,241,024 | | 257,001,472 | |

| CNN num layers | 9 | 10 | 11 | 12 | 13 | 14 |
|---|---|---|---|---|---|---|
| $5 \times 5$ filter, 3 channels | 1,725 | 1,950 | 2,175 | 2,400 | 2,625 | 2,850 |
| $5 \times 5$ filter, 128 channels | 2,873,600 | 3,283,200 | 3,692,800 | 4,102,400 | 4,512,000 | 4,921,600 |
| $5 \times 5$ filter, 1024 channels | | | | 262,195,200 | | 314,624,000 |
| $7 \times 7$ filter, 3 channels | 3,381 | 3,822 | 4,263 | 4,704 | 5,145 | 5,586 |
| $7 \times 7$ filter, 128 channels | 5,632,256 | 6,435,072 | 7,237,888 | 8,040,704 | 8,843,520 | 9,646,336 |
| $7 \times 7$ filter, 1024 channels | | | | 513,902,592 | | 616,663,040 |

| CNN num layers | 15 | 16 | 17 | 19 | 20 | 24 |
|---|---|---|---|---|---|---|
| $5 \times 5$ filter, 3 channels | 3,075 | 3,300 | 3,525 | 3,975 | 4,200 | 5,100 |
| $5 \times 5$ filter, 128 channels | 5,331,200 | 5,740,800 | 6,150,400 | 6,969,600 | 7,379,200 | 9,017,600 |
| $5 \times 5$ filter, 1024 channels | | | | | 471,910,400 | |
| $7 \times 7$ filter, 3 channels | 6,027 | 6,468 | 6,909 | 7,791 | 8,232 | 9,996 |
| $7 \times 7$ filter, 128 channels | 10,449,152 | 11,251,968 | 12,054,784 | 13,660,416 | 14,463,232 | 17,674,496 |
| $7 \times 7$ filter, 1024 channels | | | | | 924,944,384 | |

| CNN filter size | $9 \times 9$ | $13 \times 13$ | $17 \times 17$ | $25 \times 25$ | $29 \times 29$ | $57 \times 57$ |
|---|---|---|---|---|---|---|
| 5 layers, 128 channels | 4,002,048 | 8,349,952 | 14,278,912 | 30,880,000 | 41,552,128 | 160,526,592 |

Table 1: **Number of parameters of different neural network architectures used in this paper**. The number of parameters are calculated for $28 \times 28$ grayscale input / output sizes.

## M    Parameter counts for neural networks used in this paper

For convenience of cross comparing the results of different architectures with similar number of parameters, we list in Table 1 the architectures used in this paper and their associated number of parameters.

## N    Residual Networks

In this section, we evaluate training FCNs with residual connections. In particular, an identity skip connection is added for every two fully connected layers. In other words, the networks are built with two-layer blocks that computes $x \mapsto x + \text{ReLU}(W_2\text{ReLU}(W_1x))$, except that no ReLU is applied at the output layer. Adding skip connection *after* ReLU ensures the input gets passed directly to the output layer. Because the residual structure requires the same input-output shape for every block, we use 784 hidden dimensions.

Comparing Figure 41 with the vanilla FCNs in Figure 2, the identity skip connection strongly biased the FCNs towards learning the identity map. On the other hand, we can still observe that the prediction noises become stronger with larger number of hidden layers, suggesting that learning the identity function is non-trivial even with the explicit identity skip connections built into the architectures.

## O    Experiments on CIFAR-10 Images

In this section, we show experiments on some colored images from the CIFAR-10 dataset ($32 \times 32$ RGB images). Figure 42 and Figure 43 show the results from two different randomly sampled training images, respectively. The network architectures used here are nearly identical to the ones used to train on MNIST digits in the main text of the paper: the CNNs are with 128 channels and $5 \times 5$

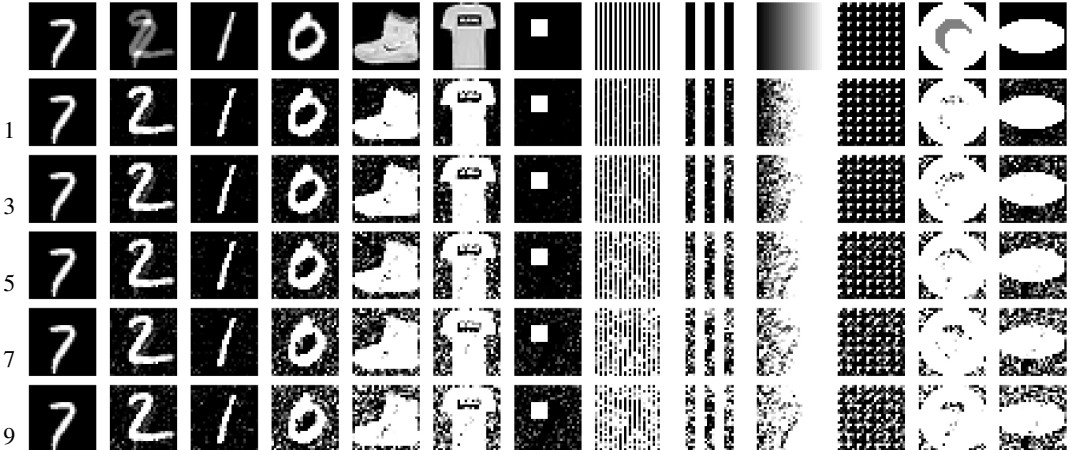

Figure 41: **Visualization of predictions from multi-layer ReLU networks with residual connections.** The first row shows the input images for evaluation, including the single training image "7" at the beginning of the row. The remaining rows show the predictions from trained multi-layer ReLU FCNs with residual connections, with the numbers on the left indicating the number of hidden layers.

filters. The FCNs are slightly modified to have larger hidden dimensions $3072 = 32\times32\times3$, to avoid explicitly enforcing bottlenecks in the hidden representations. The training loss and optimization algorithms are the same as in the MNIST case.

The main observations are consistent the results on the MNIST digits. In particular, shallow FCNs produce random noisy predictions on unseen evaluation inputs, but deep FCNs bias towards memorization and hallucinate the training image on all test outputs. For CNNs, shallow networks are capable of learning the identity function, but deep networks learn the constant function instead.

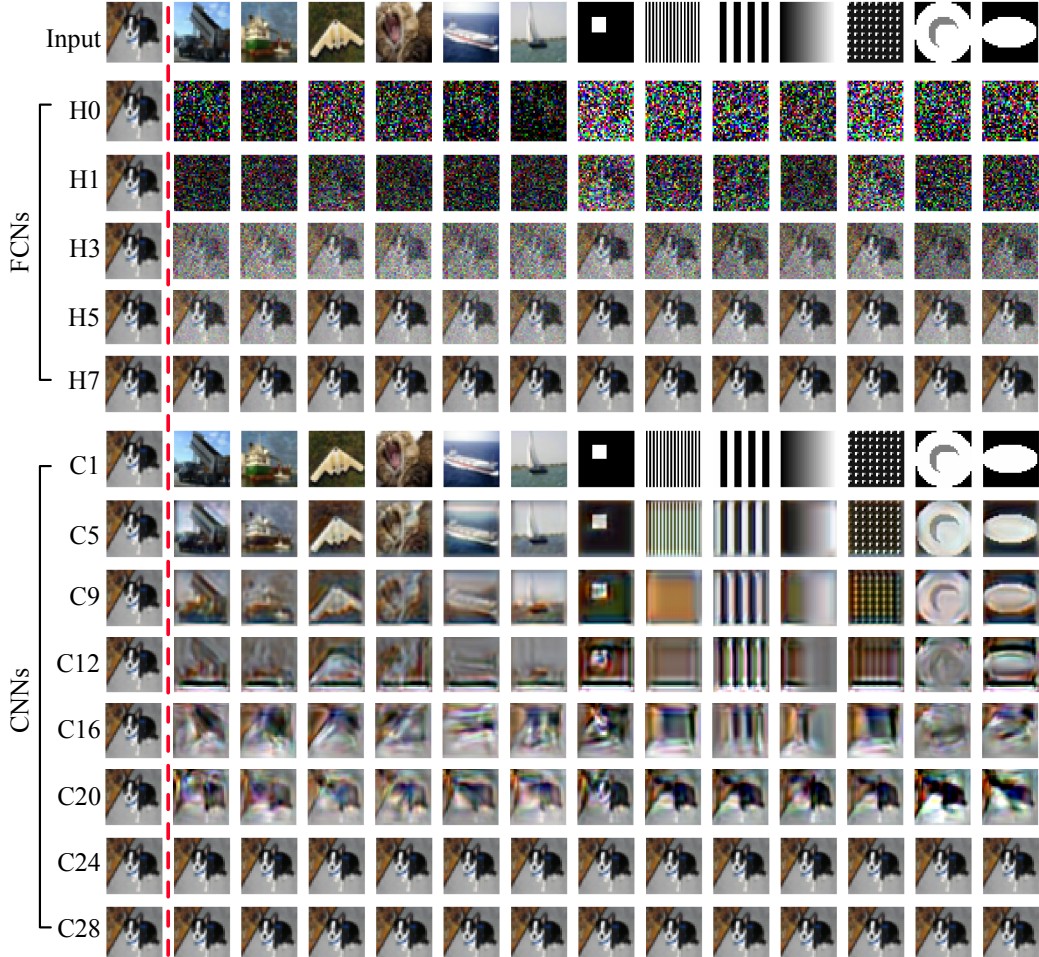

Figure 42: **Visualization of predictions from various networks trained on a single CIFAR-10 image (dog).** The first row shows the input images, where the first one is the training image and the rest are images for evaluation. The evaluation images consist of unseen images from the CIFAR-10 test set and artificially generated grayscale patterns. The grayscale patterns are duplicated into three channels before feeding into the networks. The remaining rows show the predictions from trained FCNs and CNNs. For FCNs, the numbers indicate the number of hidden layers. For example, *H3* means a 3-hidden-layer FCN. For CNNs, the numbers indicate the number of (convolutional) layers. For example, C16 means a 16-layer CNN.

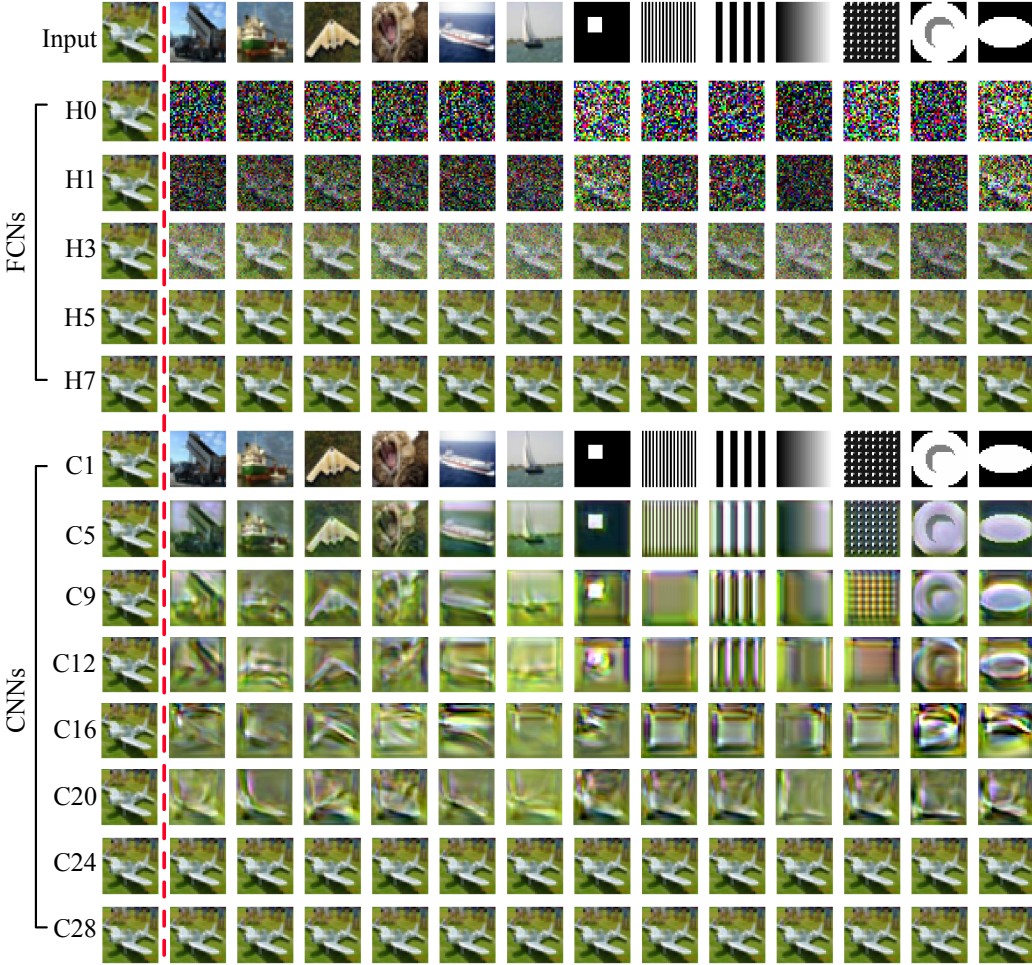

Figure 43: **Visualization of predictions from various networks trained on a single CIFAR-10 image (air plane).** The first row shows the input images, where the first one is the training image and the rest are images for evaluation. The evaluation images consist of unseen images from the CIFAR-10 test set and artificially generated grayscale patterns. The grayscale patterns are duplicated into three channels before feeding into the networks. The remaining rows show the predictions from trained FCNs and CNNs. For FCNs, the numbers indicate the number of hidden layers. For example, *H3* means a 3-hidden-layer FCN. For CNNs, the numbers indicate the number of (convolutional) layers. For example, C16 means a 16-layer CNN.

