# OpenReview forum: "Identity Crisis: Memorization and Generalization Under Extreme Overparameterization"
_ICLR.cc/2020/Conference — Accept (Poster)_

### Official Review · AnonReviewer2 · 2019-10-18
**Official Blind Review #2**

**Rating:** 6

**Review:**

This paper studies the inductive bias in deep neural networks. The authors train several FF image recognition networks and many CNN variants on a single image, and observe that most networks fall into one of two categories: either memorizing the output of the single training sample, or learning the identity function and generalizing to new, unseen images. The paper is clearly written, present a broad set of experiments, and provides interesting insights, that are somewhat surprising.

As someone from outside the area, my main concern with the paper is that I somehow missed the bigger picture. The authors present multiple different pieces of evidence that demonstrate the different conditions on which different model variants learn the different functions (memorization and generalization), but do not provide high level intuitions about what can be done with this information, and what are potential takeaways for the community. I would have expected to see some discussion after section 3, rather than jumping straight to the conclusions.

Aside from that, I think the paper puts too much of its content in the appendix (about 3 times as much content as in the main paper).

Minor:
-- section 3.2: "... if *the a* layer ..."

**Experience Assessment:**

I do not know much about this area.

**Review Assessment: Checking Correctness Of Derivations And Theory:**

I did not assess the derivations or theory.

**Review Assessment: Checking Correctness Of Experiments:**

I assessed the sensibility of the experiments.

**Review Assessment: Thoroughness In Paper Reading:**

I made a quick assessment of this paper.

---

> ### Author Response · Authors · 2019-11-12
> **Takeaways and summary**
>
> Thanks for the review. We are glad that you find our results interesting and surprising!
>
> - big picture and takeaways of the paper: there are many interesting observations from our results that could benefits the practitioners when designing models for new tasks. i) parameter counting does not strongly correlate with the generalization performance, but the structural bias of the model does. For example, being equally overparameterized, ii) training a very deep model without residual connections might be prune to memorization; while iii) adding more feature channels / dimensions is much less likely to overfit. As a result, if one is to increase the number of parameters of an existing model (maybe hoping to get smoother optimization dynamics, or to prepare for more training data), it is better to first try to increase the hidden dimension before tuning the depth, unless the nature of the data changes. Our new results, suggested by Reviewer 3, show that iv) even with explicit identity skip connections as in resnets, the amount of prediction noises still grows with the network depth.
>
> That being said, we acknowledge that the goal of this study is not primarily to identify immediate action items to help improving performances in real world problems. Instead, we aim to gain deeper understanding of the mythical inductive bias of overparameterized neural networks that seem to magically avoid overfitting and generalize well. We approach this theoretical question using empirical studies. In particular, we design a very specific and controlled experiment setting where we have clear and unambiguous definition of memorization and generalization. The extensive set of experiments are designed around this setting to reveal the inductive bias under different architectural choices. Our results show that deep networks do not always have magical inductive biases that help to generalize well. CNNs have structural biases such as local receptive field and parameter sharing that are consistent with our task, thus demonstrate stronger generalization capability than FCNs. But deeper CNNs still bias towards memorization. We believe our experiments are important contributions to the endeavor towards full understanding of the generalization behaviors of deep learning.
>
> - appendix too heavy: we apologize for referencing too often into the appendix. The main paper is self-contained and should be able to convey the main observations. However, we believe it is important to have the full details and results listed, for interested readers who would like to know more details or variations of the experiments. Some of the new experiments suggested by the reviewers (e.g. resnet, alternative initialization and alternative data) also goes into the appendix. The references into the appendix are mostly for easier indexing, but not a requirement to fully understand the paper. If you think it would improve the flow of reading, we are happy to tone down the references to the appendix, or put them in the footnotes, to make it less distracting.
>
> - typo: thanks! We fixed it in the updated version.

---

### Official Review · AnonReviewer1 · 2019-10-22
**Official Blind Review #1**

**Rating:** 3

**Review:**

This paper studies the inductive bias of neural nets by considering the toy example of learning an identity map through a single data point (and hence the NNs are always overparametrized). The authors compare CNNs versus FCNs, and find that CNNs tend to “generalize” in terms of actually learning the concept of an identity, whereas FCNs are prone to memorization. The authors also present results under various different settings such as changing the filter size or the number of hidden channels of CNNs. The conclusion is that the simpler the network architecture is, the better it generalizes. Another observation is that deep CNNs exhibit extreme memorization.

Overall, this is a well-written paper with an interesting set of experiments. However, I do have several concerns regarding the generality of the observed phenomenon in this paper. The first one is that the authors have chosen the comparison between CNNs and FCNs. In this case, I think a more fair comparison is to restrict the number of links (number of nonzero entries in the weight matrices) for both to be the same. In the current setup, however, the authors seem to consider the same number of hidden neurons, which naturally grants advantages to CNNs as they are of lower complexity.

Second, the training data is always a simple image from MNIST, and I am unsure how much this setting can generalize to other tasks, such as different data (say, music or text) or more complicated images (how do these experiments compare when we use a single data from ImageNet or CIFAR to train?). For instance, since CNNs are designed to capture invariances in natural images, it is unsurprising that they can generalize better on image data, but it would be quite astonishing if the same still holds true for acoustic data. In that case, the conclusion of this paper can be strengthened from “CNNs generalize better on image data” to “CNN generalize better”. Given the scope of the current paper, however, the best we can conclude is that “CNNs generalize better on MNIST”.

Last, again regarding the fair comparison, when comparing deep CNNs versus shallow ones, it is also of interest to see that, when restricted to the same number of parameters, if the deep CNNs still exhibit worse generalization. Otherwise, if some network complexities keep growing, we cannot really tell whether it is the network architecture that induces the inductive bias or it is simply the effect of complexities.

I also would like to suggest a future direction based on the idea in this paper: Comparing the inductive bias of GD versus SGD, a subject of intense study in the current literature. Since the authors considered a single training data, the results in this paper are always for GD. Now, say let us use 5 data, and compare the training of CNNs with different batch-size. Do the results differ? I think such a thought experiment would shed some light on the mysterious behaviors of the first-order algorithms that are widely used in practice.

Finally, a question: another submission to ICLR2020 [1] seems to suggest that optimization methods do not play a role in generalization, which is the opposite observation of this paper. Do the authors have any insight towards this contradiction?

[1] Fantastic Generalization Measures and Where to Find Them https://openreview.net/forum?id=SJgIPJBFvH

**Experience Assessment:**

I have published one or two papers in this area.

**Review Assessment: Checking Correctness Of Derivations And Theory:**

I carefully checked the derivations and theory.

**Review Assessment: Checking Correctness Of Experiments:**

I carefully checked the experiments.

**Review Assessment: Thoroughness In Paper Reading:**

I read the paper thoroughly.

---

> ### Author Response · Authors · 2019-11-12
> **Fair comparisons and CIFAR results**
>
> We are glad that you find our results interesting! Thanks for the detailed comments! We updated our paper according to the suggestions and clarify each comment below:
>
> *CNNs naturally have advantage over FCNs due to sparse and fewer number of connections*: We are primarily studying the comparisons of different architecture choices within each of the FCNs and CNNs family. But a fair comparison between FCNs and CNNs that controls the natural advantage of sparser connections in CNN could indeed be derived from our data. In particular, we have added a table summarizing the parameter counts for all the architectures used in this paper (see Table 1 in the appendix of the updated paper). From the table, we can see that a 6-layer FCN contains 3.6M parameters, while a 5-layer CNN with 5x5 filters of 1024 channels contains 78M parameters, an order of magnitude more than the FCN. Yet the CNN learns the identity function (Fig. 10a) while the FCN does not (Fig. 2). A more similar comparison is with 6 or 7-layer CNNs with 7x7 filters of 128 channels, with 3.2M and 4.0M parameters, respectively. The conclusion is the same. This is consistent with recent observations in the deep learning community that overparameterization, especially when measured by parameter counting, is not necessarily at odds with generalization.
>
> *Deep CNN vs shallow CNN should also be compared by controlling the number of parameters*: this is similar to the suggestion above, and we refer to the newly added Table 1 in the updated paper again. From the table, we can see that a 5-layer CNN with 1024 channels contains 78M parameters, while the 20-layer CNN with 3 channels contains only 4K parameters. Yet the 5-layer CNN learns the identity function while the 20-layer CNN does not.
>
> *Different data*: We added experiment results on CIFAR-10 images to the updated version of the paper (please see Appendix O). The observations are consistent with the MNIST case. Since the design of our task is to propagate all the input pixels to the outputs, the semantic complexity of the inputs matters less here than in classification tasks. We choose image data because it is intuitive to visualize and interpret (a primary goal of this paper). But we firmly agree that other domains of data and their associated architectures are valuable topics to study. For sequence inputs such as speech and language data, because both the structures in the data and the natures of the commonly used network architectures (e.g. RNNs and Transformers) are drastically different, the experiment framework, analysis and interpretations of memorization need to be completely redesigned. Therefore those studies are beyond the scope of this paper. However, we are very excited in delving into that direction for future work.
>
> *Relation to results in [1]*: Thanks for the pointer to [1]. It is a very interesting paper. However, we do not think the results in [1] are contradicting ours. In section 9 of [1], it is observed that the convergence speed is negatively correlated with the generalization gap. This is generally consistent with folklore observations that optimizers with adaptive learning rate (e.g. Adam) usually converge faster but generalize worse. And this is also aligned with our observations that sometimes with adaptive optimizers even shallow CNNs do not learn a clean identity function.
>
> *Future direction - GD vs SGD*: Thanks for the suggestion. Although there are some recent studies (e.g. [2]) that shows GD alone has strong inductive bias that could affect solution towards better generalization in some particular scenarios, we share the same feeling that the contrasting behaviors between GD and SGD is an important topic for future exploration.
>
> [2] Arora, Sanjeev, et al. "Implicit Regularization in Deep Matrix Factorization." arXiv preprint arXiv:1905.13655 (2019).

---

### Official Review · AnonReviewer3 · 2019-10-23
**Official Blind Review #3**

**Rating:** 8

**Review:**

The paper studies influence of different hyperparameters of neural networks: architecture, width, depth, initialization, optimizer, etc. on the generalization/memorization trade-off.

To do this, paper propose a clever trick: train a model to mimic identity function, so that output should be exactly the same, as input. This allows rich visualization and easy evaluation via correlation or MSE. Moreover, authors showed that it is possible to do the study on _single_ training example. Yet, in my opinion, the idea of using identity as training objective is even bigger contribution that the study itself.

The number of experiments is really terrific and lots of interesting observation is been made.  E.g, showing that over-parameterization by increasing width does not lead to overfitting, but the opposite is true. In the same time, it is hard for deeper networks to learn identity function regardless the width used.

I definitely vote this to be the oral presentation and even for the best paper award.

Questions:
    - Fig 14 (and similar): how it is possible, that for high train-test correlation, similarity of network outputs is high BOTH for constant prediction and identity? Or, might be I am just reading such visualization wrong. In that case, could you please make it more clear?
    - It would be interesting to see, how resnets behave in such setup.
    - It is surprising that He init was inferior to Xavier. Could you also try orthonormal init (Saxe et.al, ICLR 2014 https://arxiv.org/abs/1312.6120) and LSUV init (Mishkin and Matas, ICLR 2016 https://arxiv.org/abs/1511.06422)?


****
I would like to thank authors for the updated version. I agree with R2 that paper is relying too much on the appendix, but hope that authors could fix this somehow.
Now I am even more convinced that paper should be accepted and is award-quality.


**Experience Assessment:**

I have read many papers in this area.

**Review Assessment: Checking Correctness Of Derivations And Theory:**

I assessed the sensibility of the derivations and theory.

**Review Assessment: Checking Correctness Of Experiments:**

I carefully checked the experiments.

**Review Assessment: Thoroughness In Paper Reading:**

I read the paper thoroughly.

---

> ### Author Response · Authors · 2019-11-12
> **Clarification and results on resnets**
>
> Thanks for the positive review!
>
> - Fig 14: you are reading the visualization correctly. The reason that the similarity to both constant and identity predictions are high is because with high train-test correlation, the input domain of random test examples narrows down to smaller and smaller neighborhood around the training example. In this case, the difference between the identity and the constant functions are very small when evaluated over the restricted input domain.
>
> - resnet: This is a very interesting question! Our first guess was that resnets would trivially learn the identity function. The results, added as Appendix N in the updated paper, indeed show that with this architecture change, FCNs now start to learn the identity map. However, we found that even with this skip connection that explicitly carries the inputs all the way to the outputs, the predictions suffer from heavier noises as the depth increases.
>
> - He and other initialization schemes: He init samples weight values with larger variances. We found that this makes learning less stable than the Xavier init especially when no normalization layer (e.g. batch-norm) is used in the neural network architecture. We have added the orthogonal init from Saxe et.al. ICLR 2014 to the results (Fig. 31), and found the behaviors more stable than the He inits.

---

### Decision · Program_Chairs · 2019-12-19

**Decision:**

Accept (Poster)

**Comment:**

The paper studies the effect of various hyperparameters of neural networks including architecture, width, depth, initialization, optimizer, etc. on the generalization and memorization. The paper carries out a rather through empirical study of these phenomena. The authors also rain a model to mimic identity function which allows rich visualization and easy evaluation.  The reviewers were mostly positive but expressed concern about the general picture. One reviewer also has concerns about "generality of the observed phenomenon in this paper". The authors had a thorough response which addressed many of these concerns. My view of the paper is positive. I think the authors do a great job of carrying out careful experiments. As a result I think this is a good addition to ICLR and recommend acceptance.